# FBXL19 recruits CDK-Mediator to CpG islands of developmental genes priming them for activation during lineage commitment

Emilia Dimitrova[1], Takashi Kondo[2†], Angelika Feldmann[1†], Manabu Nakayama[3], Yoko Koseki[2], Rebecca Konietzny[4‡], Benedikt M Kessler[4], Haruhiko Koseki[2,5], Robert J Klose[1]*

[1]Department of Biochemistry, University of Oxford, Oxford, United Kingdom; [2]Laboratory for Developmental Genetics, RIKEN Center for Integrative Medical Sciences, Yokohama, Japan; [3]Department of Technology Development, Kazusa DNA Research Institute, Kisarazu, Japan; [4]Nuffield Department of Medicine, TDI Mass Spectrometry Laboratory, Target Discovery Institute, University of Oxford, Oxford, United Kingdom; [5]CREST, Japan Science and Technology Agency, Kawaguchi, Japan

*For correspondence:
rob.klose@bioch.ox.ac.uk

[†]These authors contributed equally to this work

Present address: [‡]Agilent Technologies, Waldbronn, Germany

Competing interests: The authors declare that no competing interests exist.

**Abstract** CpG islands are gene regulatory elements associated with the majority of mammalian promoters, yet how they regulate gene expression remains poorly understood. Here, we identify FBXL19 as a CpG island-binding protein in mouse embryonic stem (ES) cells and show that it associates with the CDK-Mediator complex. We discover that FBXL19 recruits CDK-Mediator to CpG island-associated promoters of non-transcribed developmental genes to prime these genes for activation during cell lineage commitment. We further show that recognition of CpG islands by FBXL19 is essential for mouse development. Together this reveals a new CpG island-centric mechanism for CDK-Mediator recruitment to developmental gene promoters in ES cells and a requirement for CDK-Mediator in priming these developmental genes for activation during cell lineage commitment.
DOI: https://doi.org/10.7554/eLife.37084.001

## Introduction

Multicellular development relies on the formation of cell-type-specific gene expression programmes that support differentiation. At the most basic level, these expression programmes are defined by cell signaling pathways that control how transcription factors bind DNA sequences in gene regulatory elements and shape RNA polymerase II (RNAPolII)-based transcription from the core gene promoter (reviewed in [*Spitz and Furlong, 2012*]). In eukaryotes, the activity of RNAPolII is also regulated by a large multi-subunit complex, Mediator, which can directly interact with transcription factors at gene regulatory elements and with RNAPolII at the gene promoter to modulate transcriptional activation. The Mediator complex functions through regulating pre-initiation complex formation and controlling how RNAPolII initiates, pauses, and elongates. Therefore, Mediator is central to achieving appropriate transcription from gene promoters (reviewed in [*Allen and Taatjes, 2015*; *Malik and Roeder, 2010*]).

In mammalian genomes, CpG dinucleotides are pervasively methylated and this epigenetically maintained DNA modification is generally associated with transcriptional inhibition, playing a central role in silencing of repetitive and parasitic DNA elements (*Klose and Bird, 2006*; *Schübeler, 2015*).

Most gene promoters, however, are embedded in short elements with elevated CpG dinucleotide content, called CpG islands, which remain free of DNA methylation (*Bird et al., 1985*; *Illingworth and Bird, 2009*; *Larsen et al., 1992*). Interestingly, mammalian cells have evolved a DNA-binding domain, called a ZF-CxxC domain, which can recognize CpG dinucleotides when they are non-methylated (*Lee et al., 2001*; *Voo et al., 2000*). This endows ZF-CxxC domain-containing proteins with the capacity to recognize and bind CpG islands throughout the genome (*Blackledge et al., 2010*; *Thomson et al., 2010*). There are 12 mammalian proteins encoding a ZF-CxxC domain, most of which are found in large chromatin modifying complexes that post-translationally modify histone proteins to regulate gene expression from CpG islands (*Long et al., 2013a*). This has led to the proposal that CpG islands may function through chromatin modification to affect transcription (*Blackledge et al., 2013*).

One of the first characterised ZF-CxxC domain-containing proteins was lysine-specific demethylase 2A (KDM2A) (*Blackledge et al., 2010*). KDM2A is a JmjC-domain-containing histone lysine demethylase that removes histone H3 lysine 36 mono- and di- methylation (H3K36me1/2) from CpG islands (*Blackledge et al., 2010*; *Tsukada et al., 2006*). Like DNA methylation, H3K36me1/me2 is found throughout the genome and is thought to be repressive to gene transcription (*Carrozza et al., 2005*; *Keogh et al., 2005*; *Peters et al., 2003*). Therefore, it was proposed that KDM2A counteracts H3K36me2-dependent transcriptional inhibition at CpG island-associated gene promoters (*Blackledge et al., 2010*). Sequence-based homology searches revealed that KDM2A has two paralogues in vertebrates, KDM2B and FBXL19 (*Katoh and Katoh, 2004*). KDM2B, like KDM2A, binds to CpG islands and can function as a histone demethylase for H3K36me1/2 via its JmjC domain (*Farcas et al., 2012*; *He et al., 2008*). However, unlike KDM2A, it physically associates with the polycomb repressive complex 1 (PRC1) to control how transcriptionally repressive polycomb chromatin domains form at a subset of CpG island-associated genes (*Blackledge et al., 2014*; *Farcas et al., 2012*; *He et al., 2013*; *Wu et al., 2013*). These observations have suggested that, despite extensive similarity between KDM2A and KDM2B and some functional redundancy in histone demethylation, individual KDM2 paralogues have evolved unique functions. FBXL19 remains the least well characterized and most divergent of the KDM2 paralogues. Unlike KDM2A and KDM2B, it lacks a JmjC domain and, therefore, does not have demethylase activity (*Long et al., 2013a*). Previous work on FBXL19 has suggested that it plays a role in a variety of cytoplasmic processes that affect cell proliferation, migration, apoptosis, TGFβ signalling, and regulation of the innate immune response (*Dong et al., 2014*; *Wei et al., 2013*; *Zhao et al., 2013*; *Zhao et al., 2012*). More recently, it has also been proposed to have a nuclear function as a CpG island-binding protein (*Lee et al., 2017*), but its role in the nucleus still remains poorly defined.

Here, we investigate the function of FBXL19 in mouse embryonic stem (ES) cells and show that it is predominantly found in the nucleus where it localizes in a ZF-CxxC-dependent manner to CpG island promoters. Biochemical purification of FBXL19 revealed an association with the CDK-containing Mediator complex and we discover that FBXL19 can target CDK-Mediator to chromatin. Conditional removal of the CpG island-binding domain of FBXL19 in ES cells leads to a reduction in the occupancy of CDK8 at CpG islands associated with inactive developmental genes. FBXL19 and CDK-Mediator appear to play an important role in priming genes for future expression, as these genes are not appropriately activated during ES cell differentiation when the CpG island-binding capacity of FBXL19 is abolished or CDK-Mediator is disrupted. Consistent with an important role for FBXL19 in supporting normal developmental gene expression, removal of the ZF-CxxC domain of FBXL19 leads to perturbed development and embryonic lethality in mice. Together, our findings uncover an interesting new mechanism by which CDK-Mediator is recruited to gene promoters in ES cells and demonstrate a requirement for FBXL19 and CDK-Mediator in priming developmental gene expression during cell lineage commitment.

## Results

### FBXL19 is enriched in the nucleus and binds CpG islands via its ZF-CxxC domain

FBXL19 shares extensive sequence similarity and domain architecture with the other KDM2 paralogues which are predominantly nuclear proteins (*Figure 1A*, [*Blackledge et al., 2010*;

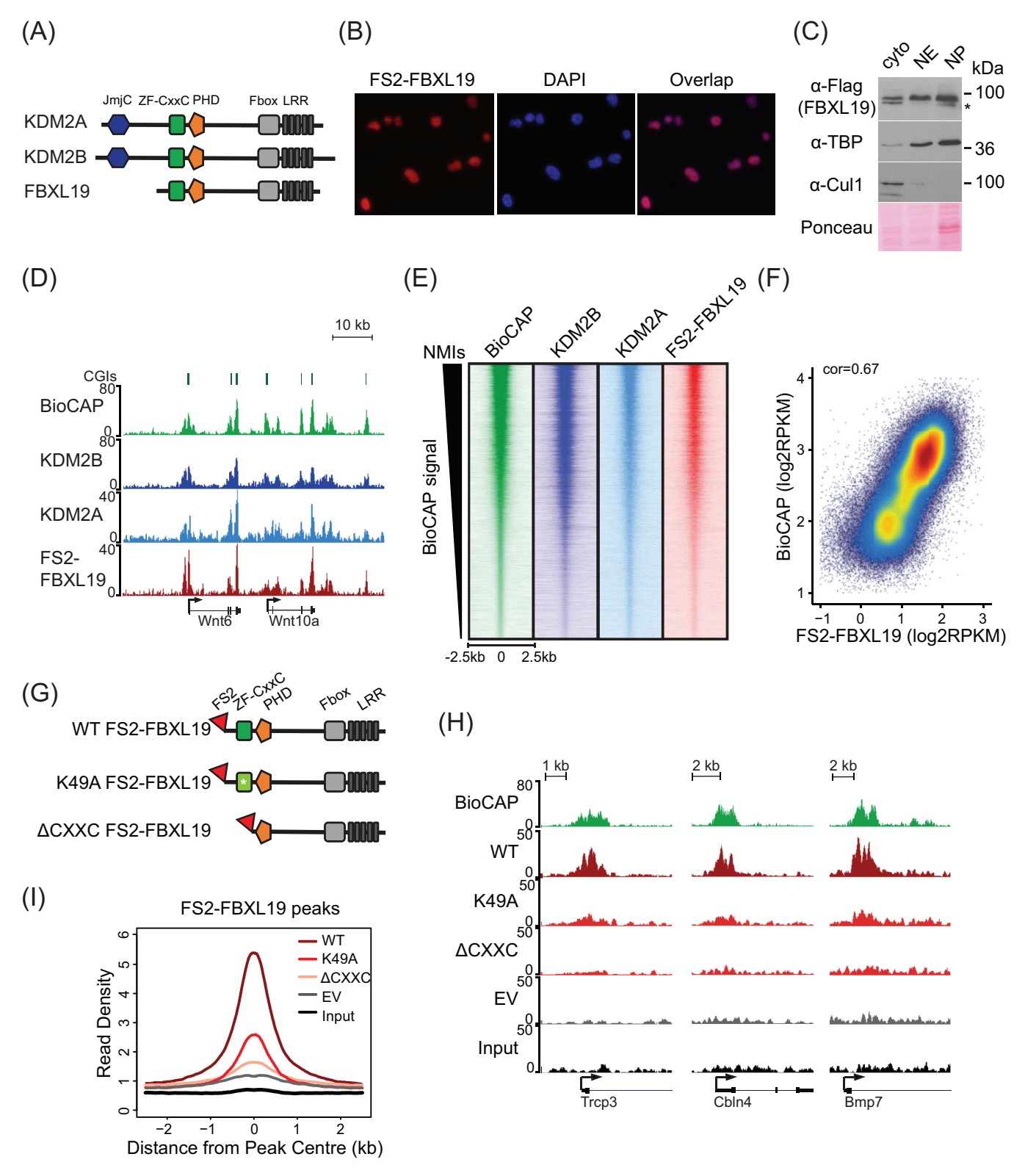

**Figure 1.** FBXL19 binds CpG islands genome-wide via its ZF-CxxC domain. (**A**) A schematic illustrating KDM2 protein domain architecture. (**B**) Immuno-fluorescent staining for FS2-FBXL19 in ES cells. (**C**) ES cell fractionation and Western blot for factors enriched in the nucleus (TBP), the cytoplasm (CUL1), and FS2-FBXL19. Ponceau S staining indicates protein loading. Cyto – cytoplasmic fraction, NE – soluble nuclear extract, NP – insoluble nuclear pellet. The asterisk indicates a non-specific band. (**D**) Screen shots showing ChIP-seq traces for KDM2B, KDM2A and FS2-FBXL19. BioCAP is included to

*Figure 1 continued on next page*

Figure 1 continued

indicate the location of non-methylated DNA and computationally predicted CpG islands (CGIs) are illustrated above. (E) Heatmaps showing enrichment of KDM2 proteins over a 5 kb region centred on non-methylated islands identified by BioCAP (NMIs) (n = 27698), sorted by decreasing BioCAP signal. BioCAP is shown for comparison. (F) A scatter plot showing the Spearman correlation between FS2-FBXL19 ChIP-seq signal and BioCAP signal at NMIs. (G) A schematic illustrating the FS2-FBXL19 transgenes. (H) A screen shot showing ChIP-seq traces for WT FS2-FBXL19 and ZF-CxxC FS2-FBXL19 mutants as in G. EV indicates empty vector control. (I) A metaplot analysis of WT FS2-FBXL19 and ZF-CxxC FS2-FBXL19 mutants over FBXL19 peaks.

DOI: https://doi.org/10.7554/eLife.37084.002

The following figure supplement is available for figure 1:

**Figure supplement 1.** FBXL19 binds CpG islands genome-wide via its ZF-CxxC domain.

DOI: https://doi.org/10.7554/eLife.37084.003

Farcas et al., 2012]), yet the function of FBXL19 has largely been described in the context of cytoplasmic processes (Dong et al., 2014; Wei et al., 2013; Zhao et al., 2013; Zhao et al., 2012). Therefore, we first examined the subcellular distribution of FBXL19 to understand whether it was unique amongst KDM2 paralogues in localizing to the cytoplasm. To achieve this, we generated a mouse ES cell line expressing epitope-tagged FBXL19 (FS2-FBXL19) and carried out immuno-fluorescent staining. This revealed that FBXL19 was almost exclusively nuclear (Figure 1B). When we carried out subcellular biochemical fractionation, FBXL19 was also enriched in the nuclear fractions in agreement with the immuno-fluorescent staining (Figure 1C). Importantly, FBXL19 was found not only in the soluble nuclear extract but also in the insoluble nuclear pellet, which contains chromatin-bound factors (Figure 1C). This suggested that FBXL19 may associate with chromatin, like other ZF-CxxC domain-containing proteins.

Based on the nuclear localization of FBXL19 and the fact that it encodes a highly conserved ZF-CxxC domain (Figure 1—figure supplement 1A), we set out to determine whether FBXL19 is a CpG island-binding protein. To achieve this, we carried out chromatin immunoprecipitation followed by massively parallel sequencing (ChIP-seq) for epitope-tagged FBXL19 and compared its binding profile with those we have previously generated for KDM2A and KDM2B in mouse ES cells (Blackledge et al., 2014; Farcas et al., 2012). A visual examination of FBXL19 ChIP-seq signal revealed that it was highly enriched at both computationally predicted CpG islands and genomic regions that contain BioCAP signal, an experimental measure of non-methylated DNA (Blackledge et al., 2012) (Figure 1D). We then identified all non-methylated CpG islands (NMIs) using BioCAP data (n = 27698) (Long et al., 2013b) and extended our analysis across the genome. We observed that FBXL19 ChIP-seq signal at NMIs was highly similar to that of KDM2A and KDM2B (Figure 1E). Furthermore, FBXL19 ChIP-seq signal correlated well with the density of non-methylated CpG dinucleotides in NMIs, similarly to KDM2A and KDM2B, (Figure 1F and Figure 1—figure supplement 1B,C), and almost all FBXL19-occupied sites fell within NMIs (Figure 1—figure supplement 1D). Together these findings demonstrate that FBXL19 is a nuclear protein that binds to CpG islands, an observation supported by a recent independent study (Lee et al., 2017).

KDM2A and KDM2B rely on defined residues in their ZF-CxxC domain to recognize non-methylated cytosine and bind CpG islands (Blackledge et al., 2010; Farcas et al., 2012). To determine whether the association of FBXL19 with CpG islands is dependent on this domain, we generated ES cell lines expressing either a mutant version of FBXL19, in which a key lysine residue was substituted to alanine (K49A, Figure 1—figure supplement 1A) to disrupt the recognition of non-methylated CpGs, or a truncated version of FBXL19, where the ZF-CxxC domain was deleted (ΔCXXC) (Figure 1G). Importantly, the expression levels of wild type (WT) and mutant FBXL19 transgenes were highly similar (Figure 1—figure supplement 1E). We then carried out ChIP-seq for epitope-tagged FBXL19 in these cell lines and compared the binding profiles of the mutant FBXL19 to that of WT FBXL19 (Figure 1H). This revealed a near complete loss of FBXL19 binding to chromatin when the ZF-CxxC domain was deleted and a slightly less dramatic effect in the K49A mutant (Figure 1H,I, and Figure 1—figure supplement 1F,G). Together, these observations demonstrate that binding of FBXL19 to CpG islands relies on an intact and functional ZF-CxxC domain.

## FBXL19 interacts with the CDK-Mediator complex in ES cells

Our ChIP-seq analyses demonstrated that FBXL19 is targeted to CpG islands in a manner that is highly similar to KDM2A and KDM2B (*Figure 1*). Although KDM2A and KDM2B localise to the same genomic regions and show a high degree of sequence conservation, they associate with different proteins (*Farcas et al., 2012*). This raised the interesting possibility that FBXL19 might also have unique interaction partners. To investigate this, we affinity-purified epitope-tagged FBXL19 from ES cell nuclear extract and identified associated proteins by mass spectrometry (AP-MS) (*Figure 2A*, *Figure 2—figure supplement 1A*, and *Figure 2—source data 1*). This revealed that FBXL19 interacts with SKP1, a known F-box-binding protein that also associates with KDM2A and KDM2B (*Bai et al., 1996*; *Farcas et al., 2012*), and the nuclear proteasome activator PSME3 (*Wójcik et al., 1998*). Interestingly, we also identified multiple subunits of the Mediator complex that appeared to interact with FBXL19 in a sub-stoichiometric manner (*Figure 2A,B*). Biochemical purifications of Mediator have identified two distinct assemblies (*Liu et al., 2001*; *Mittler et al., 2001*; *Taatjes et al., 2002*). The first is characterized by the presence of the MED26 subunit which associates with the middle region of Mediator and this form of the complex interacts with RNAPolII (*Näär et al., 2002*; *Paoletti et al., 2006*; *Sato et al., 2004*; *Takahashi et al., 2011*). Alternatively, a kinase module, composed of CDK8/CDK19, MED12/12L, MED13/13L, and CCNC, can bind to Mediator in a manner which is mutually exclusive with MED26 (*Taatjes et al., 2002*). Interestingly, our FBXL19 purification identified subunits of the kinase-containing Mediator complex (CDK-Mediator), but not MED26 or RNAPolII (*Figure 2*). This suggests that FBXL19 interacts preferentially with CDK-

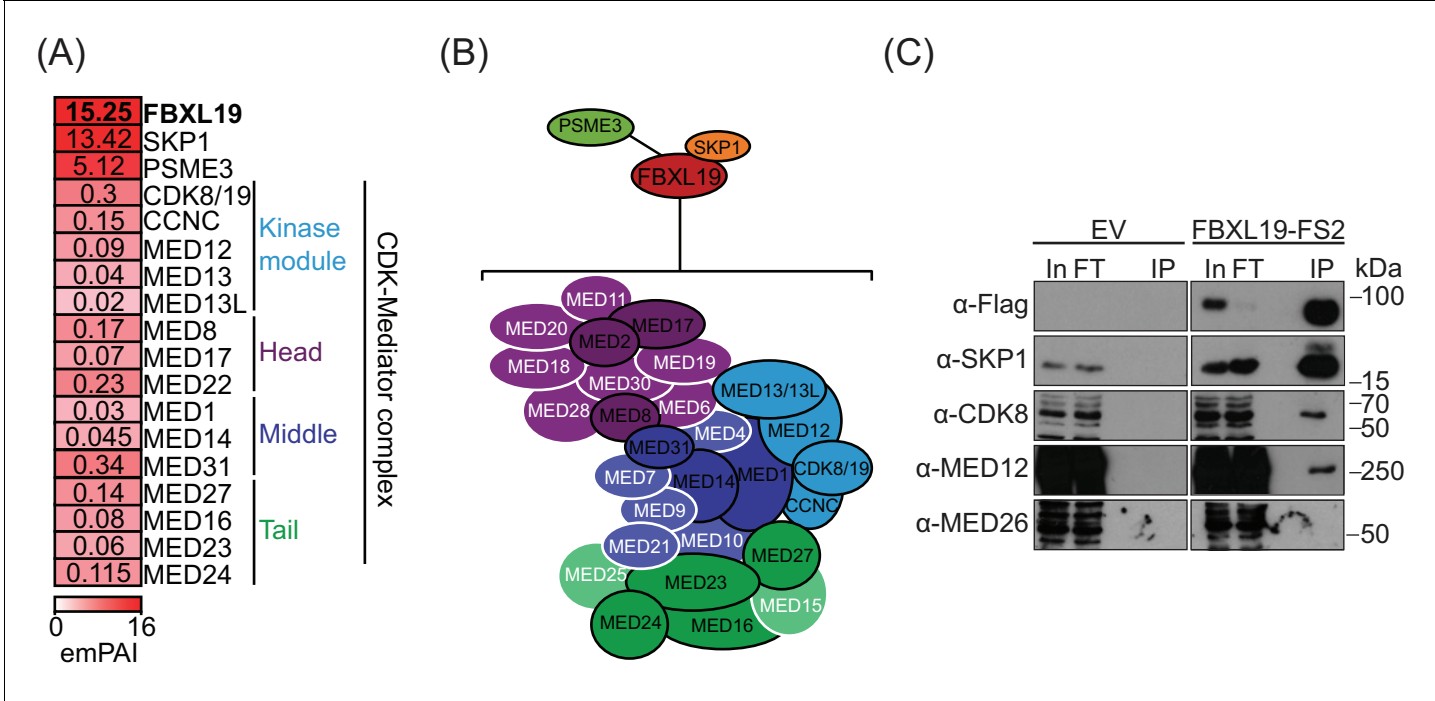

**Figure 2.** FBXL19 interacts with the CDK-Mediator complex in ES cells. (A) A heatmap representing the empirically modified protein abundance index (emPAI) of identified FBXL19-interacting proteins by affinity purification mass spectrometry (AP-MS). Data shown is an average of two biological replicates. The location of the identified Mediator components within the holocomplex is summarised on the right and in (B). (B) A schematic representation of the identified FBXL19-interacting proteins. Subunits of the CDK-Mediator complex not identified by AP-MS are shown in white. (C) Western blot analysis of FS2-FBXL19 and control purifications (EV) probed with the indicated antibodies.

DOI: https://doi.org/10.7554/eLife.37084.004

The following source data and figure supplement are available for figure 2:

**Source data 1.** Mass spectrometry data.
DOI: https://doi.org/10.7554/eLife.37084.006

**Figure supplement 1.** FBXL19 interacts with the CDK-Mediator complex in ES cells.
DOI: https://doi.org/10.7554/eLife.37084.005

Mediator, however, we cannot exclude the possibly it may also interact with MED26-Mediator. CDK8 and its paralogue CDK19 share 77% amino acid identity (89% similarity) (*Audetat et al., 2017*; *Galbraith et al., 2013*; *Sato et al., 2004*; *Tsutsui et al., 2008*) and four out of the five peptides identified by mass spectrometry were common between the two proteins (data not shown). Therefore, it is likely that FBXL19 is able to interact with both CDK8- and CDK19-containing Mediator complexes. Importantly, interaction with CDK-Mediator was also evident when we performed affinity-purification of endogenous FBXL19 (*Figure 2—figure supplement 1B–D*). However, reciprocal immunopurifications of MED12 and CDK8 failed to yield detectable FBXL19 by western blot (*Figure 2—figure supplement 1D*). This is in agreement with our mass spectrometry analysis, which indicated that the association of FBXL19 with CDK-Mediator is sub-stoichiometric, and suggests that this interaction is likely weak, as opposed to stable, inside cells.

We next wanted to determine which region of FBXL19 is required for interaction with CDK-Mediator. To do so, we transiently expressed full length FBXL19 or versions of FBXL19 with individual domains removed and performed affinity purification followed by western blot analysis. Intact FBXL19 and a version with the ZF-CxxC domain removed interacted with CDK-Mediator, whereas removing the F-box domain resulted in a loss of this interaction (*Figure 2—figure supplement 1E*). Therefore, FBXL19 relies on its F-box, and not its capacity to bind non-methylated DNA, for its association with CDK-Mediator.

Based on a candidate approach, it was recently reported that FBXL19 could interact the RNF20/40 E3 ubiquitin ligase in ES cells and regulate histone H2B lysine 120 ubiquitylation (H2BK120ub1) (*Lee et al., 2017*). In our unbiased biochemical purification of FBXL19, we did not identify an interaction with RNF20/40 by AP-MS or by western blot analysis (*Figure 2A* and *Figure 2—figure supplement 1F*). Furthermore, we failed to observe any relationship between the ability of FBXL19 to associate with CpG islands and the levels of H2BK120ub1 (*Figure 2—figure supplement 1G*). Therefore, the relevance of this proposed interaction remains unclear.

FBXL19 contains conserved F-box and leucine-rich repeat (LRR) domains. F-box proteins are known to function as scaffolds and substrate recognition modules for the SKP1-Cullin-F-box (SCF) protein complexes that ubiquitylate proteins for degradation by the proteasome (*Skaar et al., 2014*; *Skowyra et al., 1997*). Based on the association of FBXL19 with SKP1 and PSME3, we speculated that FBXL19 might function as a SCF substrate-selector for CDK-Mediator, as has previously been observed for the related F-box-containing protein FBW7 (*Davis et al., 2013*). Interestingly, however, in our FBXL19 purifications, we did not detect the SCF complex components CUL1 and RBX1/2, which are required for ubiquitin E3 ligase activity (*Skaar et al., 2014*) (*Figure 2A*). Nevertheless, we investigated in more detail whether FBXL19 might regulate CDK-Mediator protein levels. Treatment of ES cells with the proteasome inhibitor MG132, which sequesters SCF substrates on their substrate selector, did not lead to elevated levels of Mediator subunits in FBXL19 purifications as identified by AP-MS (unpublished observation). In addition, transient overexpression of FBXL19 did not cause an appreciable reduction in CDK-Mediator protein (*Figure 2—figure supplement 1H*). Based on these observations, we conclude that FBXL19 does not function as a SCF substrate selector for CDK-Mediator. This raised the interesting possibility that FBXL19 may function in a proteasome-independent manner with CDK-Mediator at CpG islands.

## FBXL19 recruits CDK-Mediator to chromatin

It has previously been shown that transcription factors can recruit the Mediator complex to enhancers and gene promoters (*Poss et al., 2013*), yet the complement of mechanisms by which Mediator is targeted to chromatin remains very poorly defined. Given that FBXL19 does not appear to regulate CDK-Mediator protein levels, we hypothesized that it might instead function to recruit CDK-Mediator to chromatin. To test this possibility, we took advantage of a synthetic system we have developed to nucleate proteins *de novo* on chromatin and test their capacity to recruit additional factors (*Blackledge et al., 2014*). Fusion of the Tet repressor DNA-binding domain (TetR) to FBXL19 allowed the recruitment of FBXL19 to a short array of Tet repressor DNA binding sites (TetO), engineered into a single site on mouse chromosome 8 ([*Blackledge et al., 2014*], *Figure 3A*). To determine whether FBXL19 was sufficient to recruit CDK-Mediator to chromatin, we stably expressed TetR or the TetR-FBXL19 fusion protein in the TetO array-containing ES cell line (*Figure 3—figure supplement 1A*). We then carried out ChIP for CDK-Mediator subunits (*Figure 3B*). Consistent with our biochemical observations (*Figure 2*), when we examined the

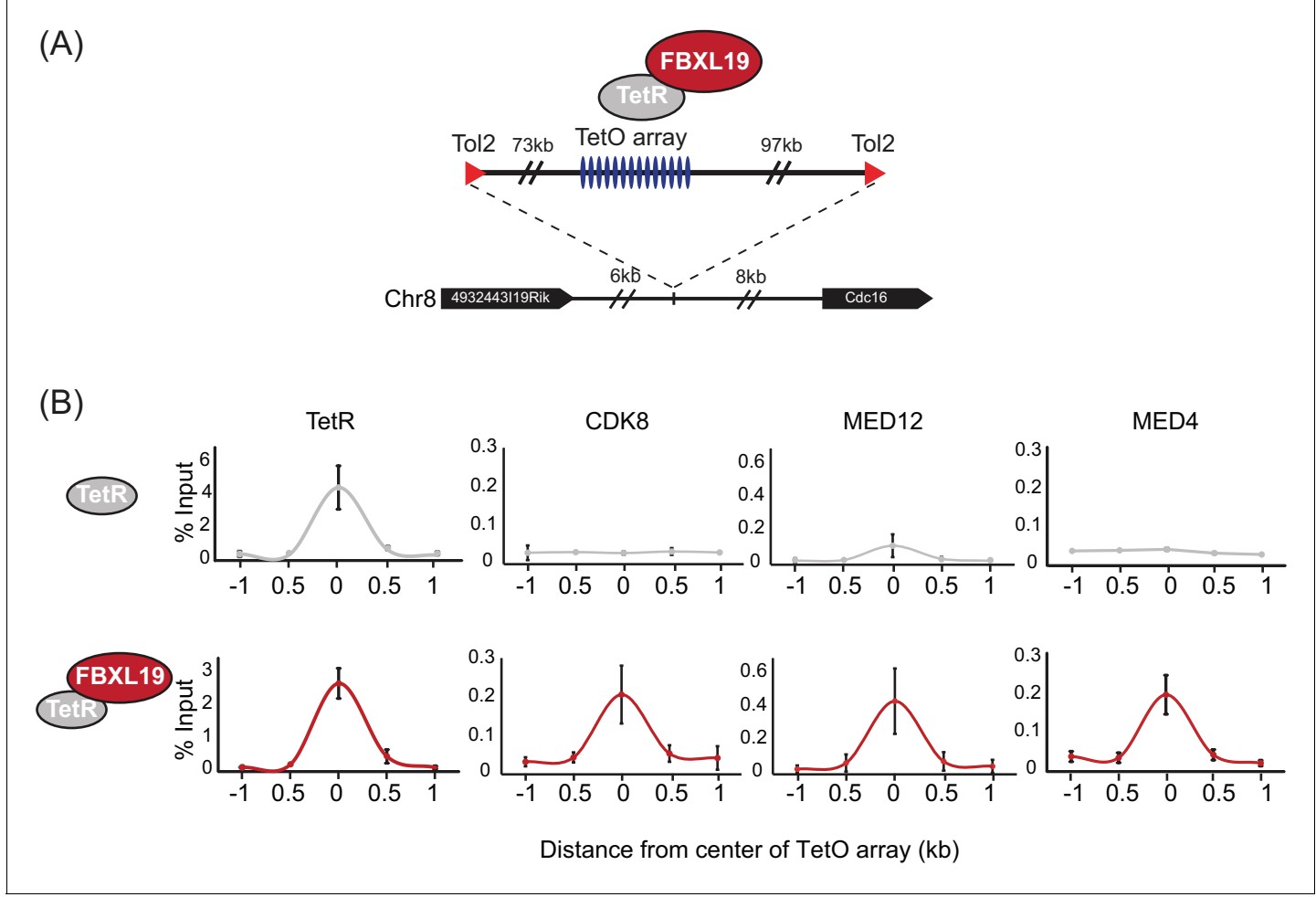

**Figure 3.** FBXL19 recruits CDK-Mediator to chromatin. (**A**) A schematic representation of the integration site of the TetO array. (**B**) ChIP-qPCR analysis of the binding of the indicated proteins across the TetO array in TetR only (top) and TetR-FBXL19 (bottom) ES cell lines. The x-axis indicates the spatial arrangement of the qPCR amplicons with respect to the center of the TetO array (in kb). Error bars represent SEM of three biological replicates.

DOI: https://doi.org/10.7554/eLife.37084.007

The following figure supplement is available for figure 3:

**Figure supplement 1.** FBXL19 recruits CDK-Mediator to chromatin.

DOI: https://doi.org/10.7554/eLife.37084.008

binding of CDK8 and MED12 over the TetO array, we observed an enrichment in the TetR-FBXL19 line. Furthermore, binding of MED4, which is part of the core Mediator complex, was also evident at the TetO array (*Figure 3B*). This suggests that FBXL19 may be sufficient to recruit a holo-CDK-Mediator complex, in keeping with its biochemical co-purification with multiple components of both the CDK module and core Mediator complex (*Figure 2*). Importantly, stable expression of TetR-KDM2A and TetR-KDM2B did not lead to CDK8 recruitment, indicating that this activity is unique to FBXL19 (*Figure 3—figure supplement 1B*). Further work will be required to determine the dynamics with which individual Mediator components are recruited to chromatin by FBXL19 as well as the precise composition of such complexes. Together these observations demonstrate that FBXL19 interacts specifically with the CDK-Mediator and can recruit this complex *de novo* to chromatin.

## FBXL19 is required for appropriate CDK8 occupancy at a subset of CpG island-associated promoters

FBXL19 binds specifically to CpG islands (*Figure 1*) and can recruit CDK-Mediator to an artificial binding site on chromatin (*Figure 3*). This raised the interesting possibility that FBXL19 may help to

recruit CDK-Mediator to CpG islands in ES cells. To examine this, we first performed CDK8 ChIP-seq in ES cells and compared it to FBXL19 ChIP-seq. Unlike FBXL19, CDK8 occupancy was not restricted to CpG islands (*Figure 4—figure supplement 1A*), with only 67.5% of CDK8 peaks overlapping with NMIs and CDK8 binding showing a limited correlation with BioCAP signal (Spearman correlation – 0.48, *Figure 4—figure supplement 1B*). This is in line with previous studies demonstrating that the Mediator complex is recruited to both enhancers and gene promoters (*Kagey et al., 2010*; *Malik and Roeder, 2010*). Interestingly, however, we observed enrichment of CDK8 at FBXL19 peaks (*Figure 4—figure supplement 1C*) with 89.4% of FBXL19 peaks overlapping with CDK8 peaks and NMIs (*Figure 4—figure supplement 1D*) raising the possibility that FBXL19 may contribute to its occupancy at these sites. To directly test this, we developed an ES cell system in which the exon encoding the ZF-CxxC domain of FBXL19 is flanked by loxP sites (*Fbxl19$^{fl/fl}$*) (*Figure 4—figure supplement 1E*) and which expresses tamoxifen-inducible ERT2-Cre recombinase. Upon addition of tamoxifen (OHT), the ZF-CxxC-encoding exon is excised, yielding a form of FBXL19 that lacks the ZF-CxxC domain (FBXL19$^{\Delta CXXC}$) (*Figure 4A,B*, and *Figure 4—figure supplement 1F*) and can, therefore, no longer bind CpG islands (*Figure 1*). This model cell system allows us to specifically examine the CpG island-associated functions of FBXL19 without affecting its other proposed roles (*Dong et al., 2014*; *Wei et al., 2013*; *Zhao et al., 2013*; *Zhao et al., 2012*). Following removal of the ZF-CxxC domain of FBXL19, we observed some reductions in FBXL19 protein levels (*Figure 4B*), but importantly CDK8 levels were unaffected (*Figure 4—figure supplement 1G*). Using this conditional *Fbxl19$^{\Delta CXXC}$* ES cell line, we then examined CDK8 occupancy on chromatin by ChIP-seq before and after OHT treatment. Genome-wide profiling of CDK8 in *Fbxll19$^{\Delta CXXC}$* ES cells did not reveal widespread alterations in CDK8 binding (*Figure 4C* left and *Figure 4—figure supplement 1H*), indicating that FBXL19 is not the central determinant driving CDK8 recruitment to most of its binding sites in the genome. Intriguingly, however, a more detailed site-specific analysis revealed a subset of CDK8 target sites that displayed significant alteration in CDK8 occupancy (*Figure 4C,D*, and *Figure 4—figure supplement 1H*). In keeping with a potential role for FBXL19 in CDK8 recruitment, the majority of the affected sites showed reduced CDK8 binding (n = 783) (*Figure 4C*).

Upon closer examination of sites with reduced CDK8 occupancy in *Fbxll19$^{\Delta CXXC}$* ES cells, we discovered that they tended to coincide with broad regions of CDK8 enrichment (*Figure 4E,F*), a feature often associated with super-enhancers (*Whyte et al., 2013*). However, a comparison of these sites to the location of super-enhancers in mouse ES cells indicated that these were distinct (*Figure 4G*). Instead, sites displaying reduction in CDK8 occupancy coincided with broad regions of non-methylated DNA (*Figure 4D,H*, and *Figure 4—figure supplement 1H*), tended to be associated with gene promoters (*Figure 4—figure supplement 1I*), and had elevated levels of FBXL19 (*Figure 4I* and *Figure 4—figure supplement 1H,I,J*). Together, these observations suggest that although binding of CDK8 to most of its target sites in the genome is achieved independently of FBXL19, a subset of broad CpG island-associated gene promoters appear to rely on FBXL19 for appropriate CDK8 occupancy.

## FBXL19 targets CDK8 to promoters of silent developmental genes in ES cells

Despite associating widely with CpG island gene promoters, FBXL19 appears to play a very specific role in maintaining appropriate CDK8 occupancy at a particular subset of broad CpG island-associated promoters (*Figure 4*). We were therefore curious to know whether something distinguishes these gene promoters from other FBXL19-bound promoters, where FBXL19 does not contribute appreciably to CDK8 occupancy (*Figure 4—figure supplement 1H*). To achieve this, we initially performed gene ontology (GO) analysis (*Huang et al., 2009*) on the genes with reduced CDK8 levels at their promoters in *Fbxl19$^{\Delta CXXC}$* ES cells. Interestingly, this revealed that these genes were strongly enriched for developmental processes (*Figure 5A*) in agreement with our previous discovery that broad CpG islands are an evolutionary conserved feature of developmentally regulated genes (*Long et al., 2013b*). In contrast, genes associated with an increased binding of CDK8 did not show any significant GO term enrichment (unpublished observation). GO analysis of the genes associated with unchanged CDK8 levels revealed terms for a broad range of basic molecular processes in line with a general role for Mediator in transcriptional regulation (*Figure 5—figure supplement 1E*).

In pluripotent mouse ES cells, many developmental genes are inactive and only become expressed as cells commit to more differentiated lineages (*Boyer et al., 2006*). Importantly, the

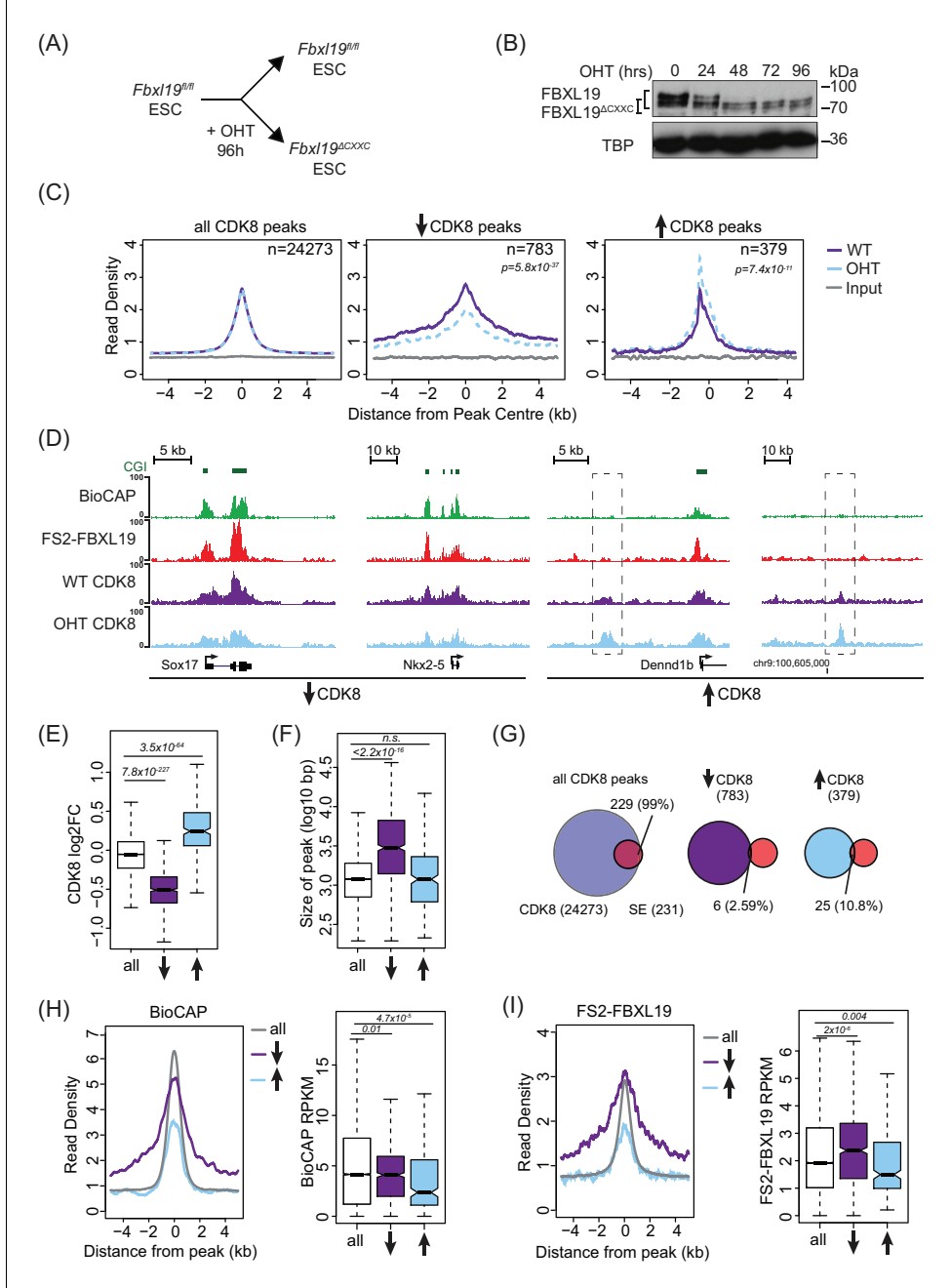

**Figure 4.** FBXL19 is required for appropriate CDK8 occupancy at a subset of CpG island promoters. (**A**) A schematic illustrating how addition of OHT (4-hydroxytamoxifen) leads to the generation of *Fbxl19^ΔCXXC* ES cells. (**B**) Western blot analysis of an OHT-treatment time course in the *Fbxl19^fl/fl* ES cell line. Hours of treatment are shown. FBXL19 protein was C-terminally tagged with a T7 epitope (*Figure 2—figure supplement 1B, C*) to allow for Western blot detection with an anti-T7 antibody. FBXL19 runs as a doublet that shifts in size following deletion of the CxxC domain. TBP was probed as a loading control. (**C**) Metaplots showing CDK8 enrichment in *Fbxl19^fl/fl* ES cells (WT) and *Fbxl19^ΔCXXC* ES cells (OHT) at all CDK8 peaks (left), peaks showing decreased CDK8 occupancy (↑, middle) and peaks showing increased CDK8 occupancy (↓, right). The number of total peaks in each group is indicated. p-values denote statistical significance calculated by Wilcoxon rank sum test comparing ChIP-seq read counts between WT and OHT samples across a 1.5-kbp interval flanking the center of CDK8 peaks. (**D**) Screen shots showing ChIP-seq traces for CDK8 in wild type *Fbxl19^fl/fl* ES cells (WT) and *Fbxl19^ΔCXXC* ES cells (OHT). BioCAP and FS2-FBXL19 tracks are given for comparison. Left – CDK8 peaks showing reduced CDK8 binding in *Fbxl19^ΔCXXC* ES cells; Right – CDK8 peaks showing increased CDK8 binding in *Fbxl19^ΔCXXC* ES cells (indicated with rectangles). (**E**) Boxplots showing log2 fold change (log2FC) of CDK8 ChIP-seq signal (RPKM) at all CDK8 peaks

*Figure 4 continued on next page*

*Figure 4 continued*

(n = 24273), and those with reduced CDK8 (n = 783, ↓) and increased CDK8 (n = 379, ↑). p-values were calculated using a Wilcoxon rank sum test. (F) Boxplots showing the size of CDK8 peaks as in (E). p-values were calculated using a Wilcoxon rank sum test. (G) Venn diagrams representing the overlap between CDK8 peaks and super enhancers (SE) at CDK8 peaks as in (E). Percent overlap of all SEs is indicated. (H) A metaplot (left) showing BioCAP enrichment at CDK8 peaks as in (E) and boxplot quantification (right) of BioCAP RPKM levels. p-Values calculated using Wilcoxon rank sum test are indicated. (I) A metalplot (left) showing FS2-FBXL19 enrichment at CDK8 peaks as in (E) and boxplot quantification (right) of FS2-FBXL19 RPKM levels. p-values calculated using Wilcoxon rank sum test are indicated.

DOI: https://doi.org/10.7554/eLife.37084.009

The following figure supplement is available for figure 4:

**Figure supplement 1.** FBXL19 is required for appropriate CDK8 occupancy at a subset of CpG island promoters.
DOI: https://doi.org/10.7554/eLife.37084.010

genes that exhibited reductions in CDK8 binding in *Fbxl19^ΔCXXC* cells were expressed at significantly lower levels than most other genes in ES cells (*Figure 5B*). Previous work has identified a subset of poised but inactive developmental genes in ES cells that are proposed to exist in a bivalent chromatin state characterized by the co-occurrence of histone H3 lysine 4 and 27 methylation (H3K4/K27me) (*Bernstein et al., 2006*; *Mikkelsen et al., 2007*). Therefore, we examined whether sites that rely on FBXL19 for normal CDK8 binding also corresponded to bivalent regions in ES cells. We found that these regions are enriched for H3K27me3 (*Figure 5—figure supplement 1A*) and have elevated H3K4me3 compared to other H3K27me3-modified sites that lack CDK8 (*Figure 5—figure supplement 1B*). Therefore, sites that rely on FBXL19 for normal CDK8 binding and correspond to silent developmental gene promoters are also bivalent. To ask whether these genes are induced during cell lineage commitment, we compared their expression levels in ES cells and following retinoic acid (RA) treatment which induces differentiation (*Figure 6—figure supplement 1B,C*). This clearly demonstrated that the genes which rely on FBXL19 for appropriate CDK8 binding in the ES cell state can become transcriptionally activated during stem cell lineage commitment (*Figure 5C*).

The observation that FBXL19 appears to be important for CDK8 occupancy at largely inactive genes was intriguing given that previous work characterizing Mediator function has usually focussed on its activity at actively transcribed or induced genes (reviewed in [*Poss et al., 2013*]). We therefore set out to examine the relationship between FBXL19, CDK8 and gene expression in more detail. We first separated all promoters based on expression of their associated genes (*Figure 5D* and *Figure 5—figure supplement 1D*). We observed that CDK8 occupancy was in general linked to gene activity (*Figure 5E*), in agreement with previous reports (*Alarcón et al., 2009*; *Bancerek et al., 2013*; *Donner et al., 2010*; *Donner et al., 2007*; *Fryer et al., 2004*; *Galbraith et al., 2013*). However, at FBXL19-bound gene promoters, CDK8 occupancy was similar in both the lowly and highly expressed subsets of genes (*Figure 5E*). Importantly, CDK8 occupancy at promoters of lowly expressed genes was reduced in *Fbxl19^ΔCXXC* ES cells (*Figure 5F*). Therefore, these observations reveal that CDK8 is enriched at promoters of inactive or lowly expressed genes in ES cells and its binding is dependent on recognition of CpG islands by FBXL19.

## Removing the CpG island-binding domain of FBXL19 results in a failure to induce developmental genes during ES cell differentiation

FBXL19 appears to play a role in recruiting CDK8 to a class of genes that are repressed in the ES cell state and become activated during cell linage commitment (*Figure 5*). However, it remained unclear whether FBXL19 is required for the activation of these developmental genes. To address this question, we induced differentiation of *Fbxl19^fl/fl* and *Fbxl19^ΔCXXC* ES cells with RA (*Figure 6A*) and compared the expression of several genes which showed reduced CDK8 binding in *Fbxl19^ΔCXXC* ES cells (*Figure 6B*). We observed no significant differences in the expression of these genes in wild-type *Fbxl19^fl/fl* and *Fbxl19^ΔCXXC* ES cells where these genes are silent. Strikingly, however, following RA treatment, *Fbxl19^ΔCXXC* cells failed to induce these genes appropriately (*Figure 6B*). Similarly, developmental genes were not appropriately induced during embryoid body differentiation of *Fbxl19^ΔCXXC* ES cells (*Figure 6—figure supplement 1A*). Importantly, this demonstrates that FBXL19

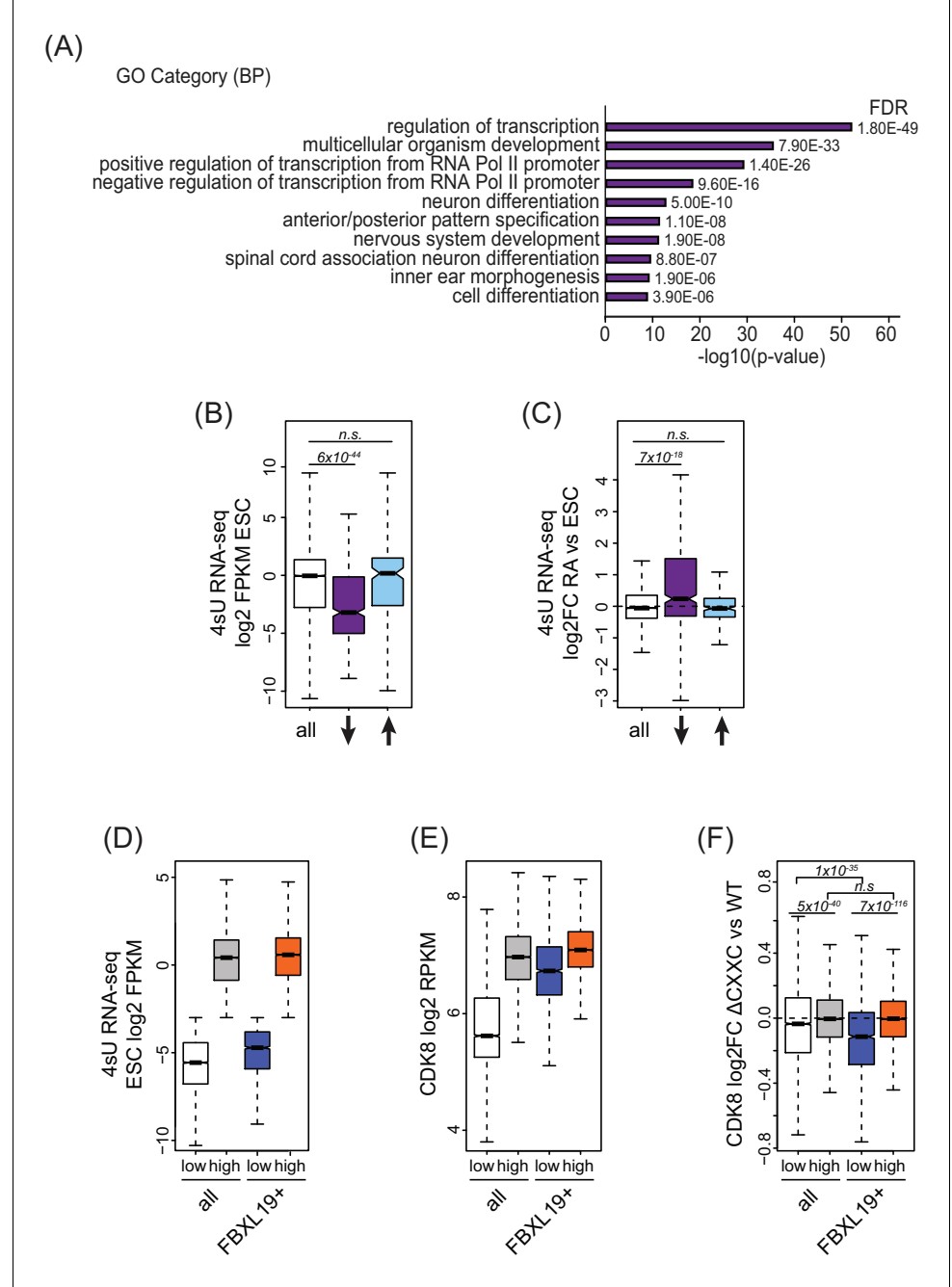

**Figure 5.** FBXL19 targets CDK8 to promoters of silent developmental genes in ES cells. (**A**) Gene ontology analysis of genes associated with a decrease in CDK8 binding (n = 673).(**B**) A boxplot showing expression levels (log2FPKM) of CDK8-bound genes in wild-type ES cells. CDK8-associated genes are divided based on CDK8 binding in $Fbxl19^{\Delta CXXC}$ ES cells: all CDK8-bound (n = 15161), reduced CDK8 (n = 673, ↓), and increased CDK8 binding (n = 255, ↑). p-values were calculated using a Wilcoxon rank sum test. (**C**) A boxplot showing the change in gene expression (log2 fold change) observed by 4sU RNA-seq of CDK8-associated genes (as in B) following RA treatment. p-values were calculated using a Wilcoxon rank sum test. (**D**) A boxplot comparing gene expression levels (log2FPKM) of all (n = 19310) and FBXL19-bound (FBXL19+, n = 11368) genes separated by low (all genes n = 7417; FBxl19+ genes n = 2031) and high expression levels (all genes n = 11893, FBxl19+ genes n = 9337) in ES cells (based on *Figure 5—figure supplement 1B*). (**E**) A boxplot showing CDK8 enrichment at all and FBXL19-bound genes separated by expression level as in (**D**). (**F**) A boxplot showing change in CDK8 binding at the TSSs of all and FBXL19-bound genes divided by expression level as in (**D**). p-values were calculated using a Wilcoxon rank sum test.

*Figure 5 continued on next page*

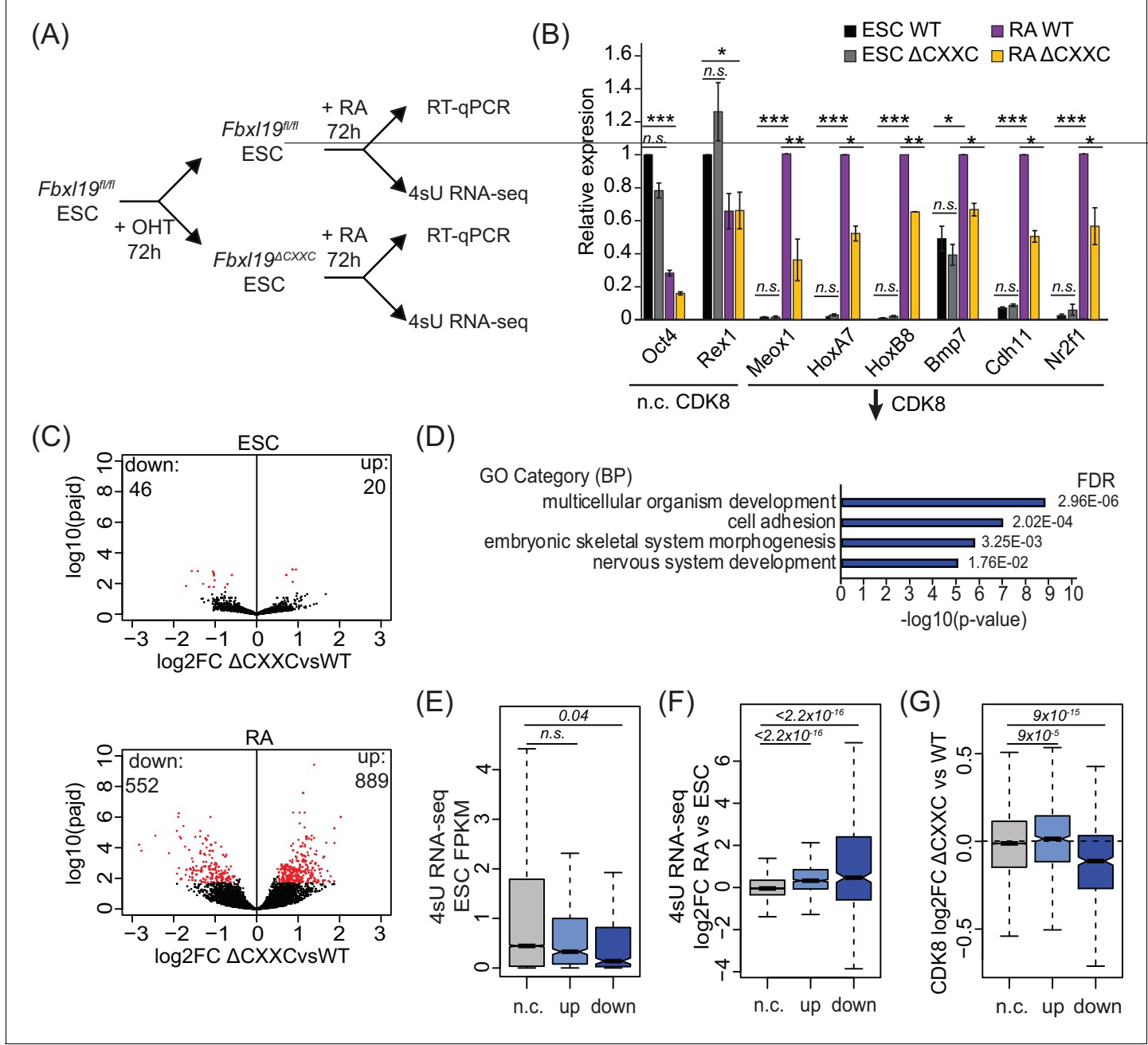

**Figure 6.** Removing the CpG island-binding domain of FBXL19 results in a failure to induce developmental genes during ES cell differentiation. (**A**) A schematic illustrating the OHT treatment and differentiation approach. (**B**) RT-qPCR gene expression analysis of genes showing decreased CDK8 binding in *Fbxl19^{ΔCXXC}* ES cells before (ESC) and after RA induction. Expression is relative to the average of two house-keeping genes. Error bars show SEM of three biological replicates, asterisks represent statistical significance calculated by Student T-test: *p<0.05, **p<0.01, ***p<0.001. (**C**) Volcano plots showing differential expression (log2 fold change) comparing WT and *Fbxl19^{ΔCXXC}* cells in the ES cell (top) and RA-induced state (bottom). Differentially expressed genes (log2FC < −0.5 or log2FC > 0.5, padj <0.1) are shown in red. The number of genes considered to be significantly altered in expression are indicated. (**D**) Gene ontology analysis of genes with decreased expression in *Fbxl19^{ΔCXXC}* ES cells following RA treatment (n = 552). (**E**) A boxplot indicating the expression level (FPKM) based on 4sU RNA-seq in wild-type ES cells of the gene groupings in (**C**): n.c. genes n = 17869 (no significant change), up genes = 889, down genes = 552. p-values calculated using a Wilcoxon rank sum test are indicated. (**F**) A boxplot indicating the log2 fold change in gene expression of the gene groupings in (**E**) upon RA differentiation of ES cells. p-values calculated using a Wilcoxon rank sum test are indicated. (**G**) A boxplot indicating the change in CDK8 binding (log2FC) at the promoters of the gene groupings in (**E**) in ES cells. p-values calculated using a Wilcoxon rank sum test are indicated.

DOI: https://doi.org/10.7554/eLife.37084.013

*Figure 6 continued on next page*

*Figure 6 continued*

The following source data and figure supplement are available for figure 6:

**Source data 1.** Differential gene expression analysis.
DOI: https://doi.org/10.7554/eLife.37084.015
**Figure supplement 1.** Removing the CpG island-binding domain of FBXL19 results in a failure to induce developmental genes during ES cell differentiation.
DOI: https://doi.org/10.7554/eLife.37084.014

is required for the appropriate activation of developmental gene expression during cell lineage commitment.

To understand the extent to which genes are not appropriately induced during differentiation of $Fbxl19^{\Delta CXXC}$ ES cells, we examined ongoing transcription in both the ES cell state and after RA-mediated differentiation using short time-scale 4-thiouridine-based labeling and RNA sequencing (4sU RNA-seq) (*Figure 6A*). We observed a significant number of genes that become induced upon RA treatment in wild-type cells (n = 4051) (*Figure 6—figure supplement 1B* and *Figure 6—source data 1*). GO term analysis confirmed that these genes are associated with ES cell differentiation and early embryonic development (*Figure 6—figure supplement 1C*). We then used differential gene expression analysis to compare transcription in $Fbxl19^{fl/fl}$ and $Fbxl19^{\Delta CXXC}$ cells (*Figure 6C* and *Figure 6—source data 1*). While very few significant changes in gene expression were observed in the ES cell state, following RA-induced differentiation we identified a large number of genes (n = 552) that had significantly lower expression levels in $Fbxl19^{\Delta CXXC}$ cells (*Figure 6C*). This is consistent with our results when examining individual genes (*Figure 6B* and *Figure 6—figure supplement 1A*) and with a recent study where the expression of a selection of genes was examined following FBXL19 knock-down (*Lee et al., 2017*). GO analysis revealed that the set of genes that were not appropriately activated were associated with genes involved in developmental processes (*Figure 6D* and *Figure 6—source data 1*), unlike genes with increased expression (n = 889) which were not associated with these processes (*Figure 6—figure supplement 1D* and *Figure 6—source data 1*). It is possible that the observed increases in gene expression in $Fbxl19^{\Delta CXXC}$ cells following RA induction result from secondary effects or as of yet unidentified roles of FBXL19 in inhibiting gene expression. Nevertheless, consistent with our analysis of FBXL19-dependent CDK8 occupancy at promoters of developmental genes (*Figure 5B,C*), genes not appropriately induced were lowly expressed in wild-type ES cells (*Figure 6E*) and tended to become activated following RA induction (*Figure 6F*). Importantly, these genes exhibited a reduction in CDK8 binding in $Fbxl19^{\Delta CXXC}$ ES cells (*Figure 6G* and *Figure 6—figure supplement 1E*), supporting the idea that FBXL19 recruits CDK8 to this subset of CpG island-associated developmental genes and that this may prime these genes for activation during differentiation.

## FBXL19 target genes rely on CDK-Mediator for activation during differentiation

Given that FBXL19 is required for CDK8 occupancy at the promoters of a series of silent developmental genes and for their activation during differentiation (*Figures 5* and *6*), we hypothesized that FBXL19 may prime these genes for future activation through the activity of CDK-Mediator. To address this interesting possibility, we developed a system to conditionally remove MED13 and its closely related paralogue MED13L in ES cells ($Med13/13l^{fl/fl}$ ERT2-Cre ES cells) (*Figure 7A* and *Figure 7—figure supplement 1A*). We chose to inactivate MED13/13L as it has previously been shown to physically link the CDK-kinase module to the core Mediator complex and underpin the formation of a functional CDK-Mediator (*Knuesel et al., 2009*; *Tsai et al., 2013*). Treatment of the $Med13/13l^{fl/fl}$ ES cells with OHT resulted in a loss of MED13 and MED13L protein (MED13/13L KO) and reduced levels of the other subunits of the CDK-kinase module (*Figure 7B*). Importantly, removal of MED13/13L also led to a loss of CDK8 binding to chromatin (*Figure 7C*) but did not have an appreciable effect on the expression of FBXL19 target genes in the ES cell state (*Figure 7D*). We next induced differentiation of the MED13/13L KO cells with RA (*Figure 7A*). Importantly, when we then analysed the expression of a series of genes that rely on FBXL19 for activation during differentiation (*Figure 7D*), we observed that these genes also failed to appropriately induce in MED13/13L KO

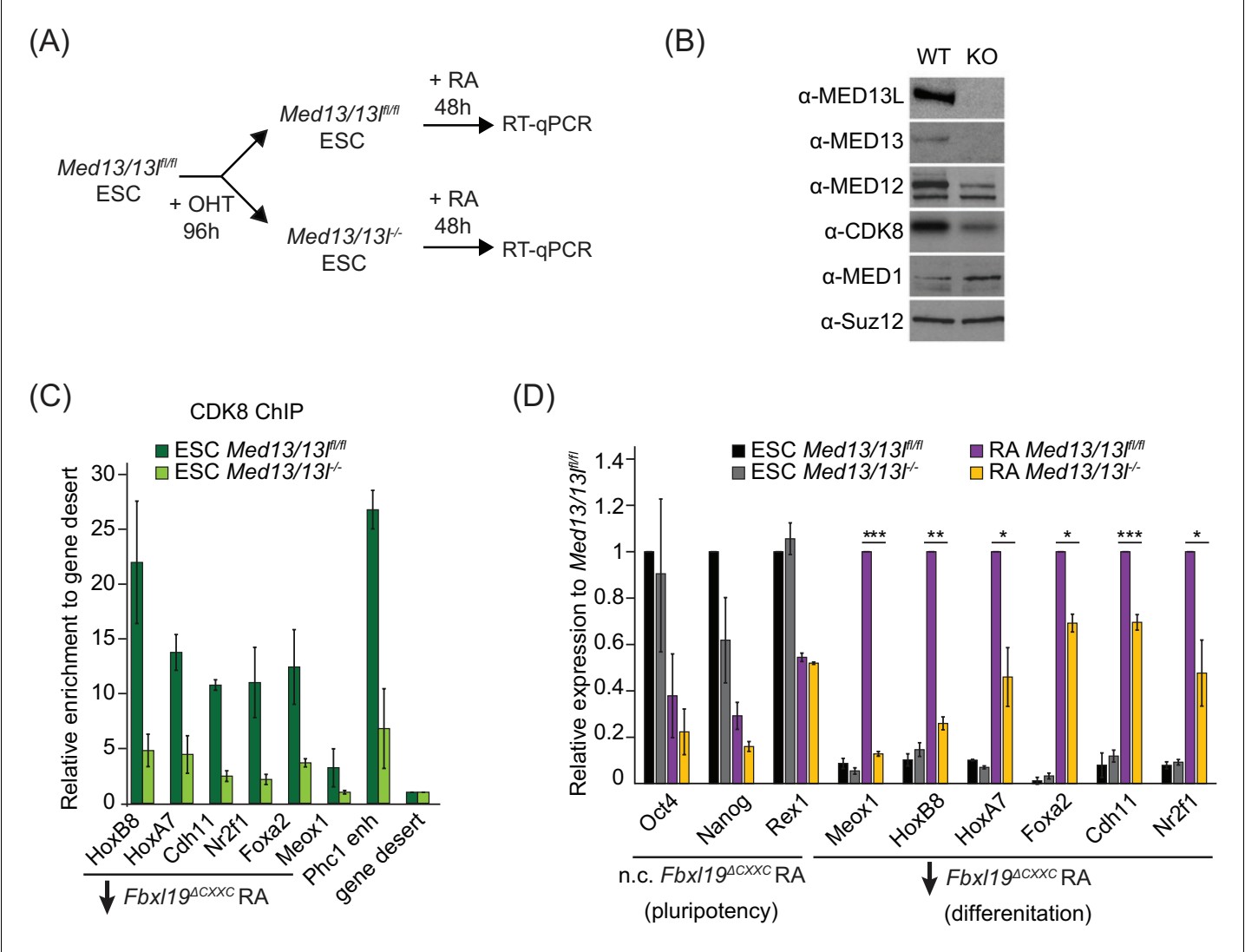

**Figure 7.** FBXL19 target genes rely on CDK-Mediator for activation during differentiation. (**A**) A schematic illustrating the OHT treatment to generate MED13/13L KO ES cells and differentiation approach. (**B**) Western blot analysis showing the efficiency of MED13/13 knock-out upon 96 hr OHT treatment of *Med13/13l*$^{fl/fl}$ ES cells. Suz12 was blotted as a loading control. (**C**) ChIP-qPCR showing CDK8 enrichment in WT and MED13/13L KO ES cells. Enrichment is relative to gene desert control region. Error bars show standard deviation of two biological replicates. (**D**) RT-qPCR gene expression analysis in WT and MED13/13L KO ES cells before and after RA induction. Expression was normalized to the expression of the PolIII-transcribed gene *tRNA-Lys* and is represented as relative to WT ES cells (for pluripotency genes) or RA-treated WT cells (for differentiation markers). Error bars show SEM of three biological replicates, asterisks represent statistical significance calculated by Student T-test: *p<0.05, **p<0.01, ***p<0.001.
DOI: https://doi.org/10.7554/eLife.37084.016

The following figure supplement is available for figure 7:

**Figure supplement 1.** FBXL19 target genes rely on CDK-Mediator for activation during differentiation.
DOI: https://doi.org/10.7554/eLife.37084.017

cells (*Figure 7D*). Together this suggests that FBXL19, via its association with CpG islands, recognises a subset of silent developmental genes in ES cells in order to recruit the CDK-Mediator complex and prime these genes for future activation.

## Ablating the capacity of FBXL19 to bind CpG islands causes embryonic lethality

Our results suggest that FBXL19 contributes to gene activation during cell lineage commitment via recognition of CpG islands and recruitment of the CDK-Mediator complex. Given that *Fbxl19*$^{\Delta CXXC}$

(A)

| Stage | WT | *Fbxl19*$^{CXXC\Delta/+}$ | *Fbxl19*$^{CXXC\Delta/\Delta}$ | Total |
|---|---|---|---|---|
| 9.5 dpc | 4 (21%) | 9 (47%) | 6 (32%) | 19 |
| 10.5 dpc | 20 (26%) | 40 (52%) | 17 (22%) | 77 |
| newborn | 8 (33%) | 16 (67%) | 0 (0%) | 24 |

(B)

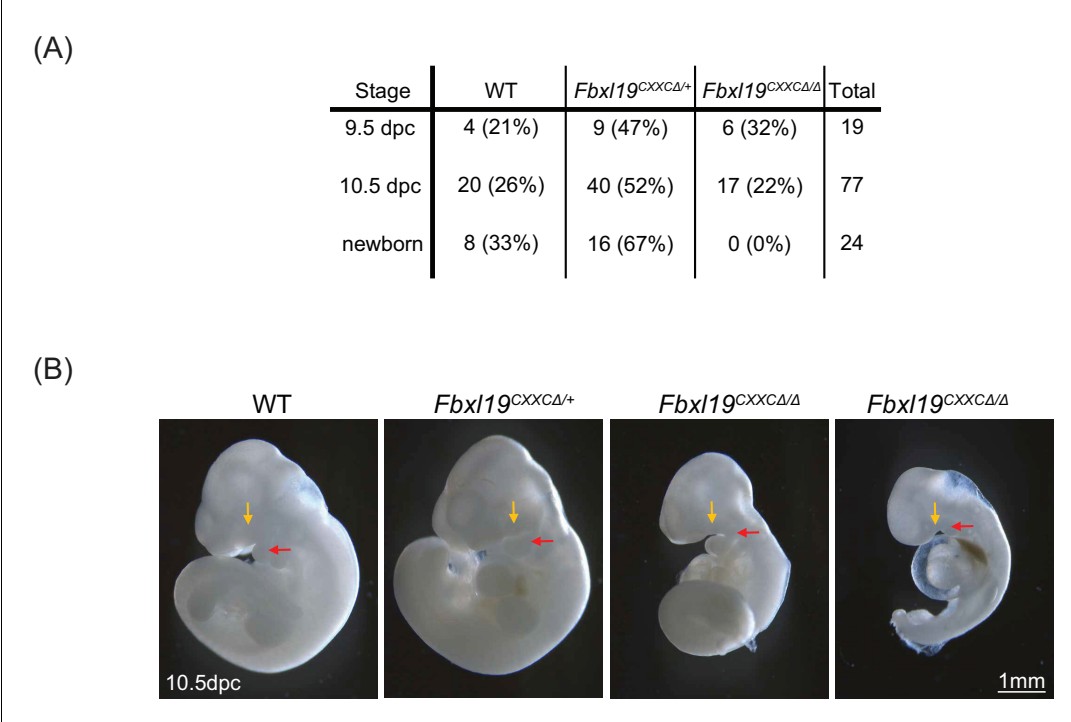

**Figure 8.** Deletion of the ZF-CxC domain of FBXL19 leads to mouse embryonic lethality. (**A**) After crossing heterozygotes, the number of live embryos (9.5 dpc or 10.5 dpc) or newborn pups with the indicated genotypes is indicated with the percentages shown in parentheses. (**B**) The morphological changes in 10.5 dpc *Fbxl19*$^{CXXC\Delta/\Delta}$ embryos are indicated. Lateral views of wild-type (wt), together with heterozygous (*Fbxl19*$^{CXXC\Delta/+}$) and homozygous (*Fbxl19*$^{CXXC\Delta/\Delta}$) mutants are shown. Maxillary and mandibular components of first branchial arches are indicated by yellow and red arrows, respectively.
DOI: https://doi.org/10.7554/eLife.37084.018

cells display impaired gene activation in our *in vitro* differentiation model, we asked whether the inability of FBXL19 to bind and function at CpG islands could also affect mouse development. In order to address this, we generated *Fbxl19*$^{\Delta CXXC}$ mutant mice by crossing *Fbxl19*$^{fl/fl}$ animals with animals constitutively expressing Cre recombinase. From these crosses, we failed to obtain any viable *Fbxl19*$^{\Delta CXXC}$ homozygous offspring, indicating that removal of the ZF-CxC domain of FBXL19 leads to embryonic lethality (*Figure 8A*). We then investigated at which stage *Fbxl19*$^{\Delta CXXC}$ homozygous embryos were affected and found homozygous embryos were observed at normal Mendelian ratios until 10.5 days postcoitum (dpc) (*Figure 8A*). At 9.5 dpc, gross embryonic morphology appeared to be intact but at 10.5 dpc (*Figure 8B*) the embryos exhibited a clear growth retardation and showed, to varying extents, reduced elongation of the trunk, hypomorphic limb buds and cephalic region. This included undeveloped facial mesenchyme, including defects in maxillary and mandibular components of first branchial arches (indicated by yellow and red arrows in *Figure 8B*, respectively), and hypomorphic cardiac mesenchyme. Some living *Fbxl19*$^{\Delta CXXC}$ homozygous embryos, characterized by a beating heart, were observed at 12.5 dpc but they were of similar size and external appearance to 10.5 dpc mutant embryos suggesting that normal development had ceased by this point. Together our findings demonstrate that FBXL19, and its ability to recognize CpG islands, is essential for normal mouse embryonic development.

## Discussion

Here, we discover that FBXL19 recognizes CpG islands throughout the genome in a ZF-CxC-dependent manner (*Figure 1*). Unlike other ZF-CxC proteins which associate with chromatin-modifying complexes, we show that FBXL19 interacts with the CDK-Mediator complex (*Figure 2*). This uncovers an unexpected link between CpG islands and a complex that regulates gene expression through interfacing with the transcriptional machinery. We demonstrate that FBXL19 can recruit

CDK-Mediator to chromatin (*Figure 3*) and, via recognition of CpG islands, plays an interesting role in supporting CDK8 occupancy at a subset of promoters associated with developmental genes, which are inactive in ES cells (*Figure 4* and *Figure 5*). At these CpG islands, FBXL19 and CDK-Mediator function to prime the associated genes for activation during ES cell differentiation (*Figure 6* and *Figure 7*). Consistent with an important role of FBXL19 in supporting normal developmental gene expression, removal of the ZF-CxxC domain of FBXL19 leads to perturbed development and embryonic lethality in mice (*Figure 8*). Together these new discoveries reveal that CpG islands and FBXL19 can interface with CDK-Mediator to orchestrate normal gene expression during lineage commitment.

Previously, another F-box protein, FBW7, was shown to function as an E3 ubiquitin ligase substrate selector for the MED13/13L subunit of CDK-Mediator and to regulate its stability through proteasomal degradation (*Davis et al., 2013*). By controlling CDK-Mediator abundance, one could envisage how this might shape Mediator function in gene expression. FBXL19 also encodes an F-box domain and associates with SKP1, a central component of SCF E3 ubiquitin ligase complexes. However, we do not find evidence that FBXL19 regulates CDK-Mediator stability via the proteasome (*Figure 2—figure supplement 1H*). Instead, we discover that FBXL19 can recruit CDK-Mediator to chromatin and, more specifically, to CpG islands via its ZF-CxxC domain to support gene activation. Although we currently do not know the defined subunits and surfaces in CDK-Mediator that FBXL19 interacts with, our biochemical experiments indicate that this relies on an intact F-box domain in FBXL19 (*Figure 2—figure supplement 1E*). It is tempting to speculate that FBXL19 could interact directly with MED13/13L given their recognition by the related F-box protein, Fbw7 (*Davis et al., 2013*). Nevertheless, a key observation in our work is that FBXL19 appears to have functionally diverged from other F-box proteins during vertebrate evolution by acquiring a DNA-binding domain that allows it to recruit proteins to chromatin, instead of targeting them for ubiquitylation. This general feature is shared with the FBXL19 paralogue, KDM2B, which associates with and recruits the PRC1 complex to CpG islands (*Farcas et al., 2012*; *He et al., 2013*; *Wu et al., 2013*). In agreement with our observations, a large-scale proteomics screen previously suggested that an interaction between FBXL19 and Mediator may also exist in cancer cells (*Tan et al., 2013*). Given that CDK8 can function as an oncogene (*Adler et al., 2012*; *Firestein et al., 2008*; *Morris et al., 2008*), it will be interesting to understand whether FBXL19 plays a role in targeting CDK-Mediator to CpG islands in non-embryonic tissues, and whether this activity may promote CDK8-driven tumorigenesis.

Individual Mediator subunits have been shown to interact with DNA-binding transcription factors that recruit the Mediator complex to specific DNA sequences in regulatory elements and support gene expression (*Black et al., 2006*; *Blazek et al., 2005*; *Fondell et al., 1996*; *Malik and Roeder, 2010*). In part, Mediator is thought to achieve this by bridging enhancer elements to the core gene promoter and RNAPolII (*Allen and Taatjes, 2015*; *Jeronimo et al., 2016*; *Petrenko et al., 2016*). However, in mammals these proposed mechanisms are extrapolated from studying only a subset of transcription factors and genes. Therefore, the defined mechanisms that shape Mediator occupancy on chromatin remain very poorly understood. Here, we provide evidence for a completely new gene promoter-associated CDK-Mediator targeting mechanism that relies on FBXL19 and CpG island recognition. Surprisingly, although FBXL19 localizes broadly to CpG islands throughout the genome (*Figure 1*), we only observed a reliance on FBXL19 for CDK8 binding at a subset of sites (*Figure 4*). Similarly, KDM2B binds broadly to CpG islands, yet has a specificity in shaping PRC1 recruitment and polycomb repressive chromatin domain formation at a subset of developmental genes in mouse ES cells (*Farcas et al., 2012*). We have previously suggested that this could result from ZF-CxxC domain-containing proteins broadly sampling CpG islands, with their activity or affinity for certain sites being shaped by local gene activity or chromatin environment (*Klose et al., 2013*). In keeping with these general ideas, FBXL19 is required for appropriate CDK8 binding at bivalent genes in ES cells (*Figure 5—figure supplement 1A,B*). In future work, it will be important to understand whether this unique chromatin state and/or the absence of transcriptional activity at these sites define the requirement for FBXL19 in CDK8 recruitment, or whether FBXL19 targeting occurs at most CpG islands sites but is masked through redundancy with transcription factor and gene activity-dependent targeting modalities.

Historically, CDK-Mediator has been associated with gene repression (*Akoulitchev et al., 2000*; *Chi et al., 2001*; *Elmlund et al., 2006*; *Hengartner et al., 1998*; *Knuesel et al., 2009*; *Pavri et al., 2005*). Therefore, we were not surprised to observe that CDK8 occupied the promoters of repressed

developmental genes in ES cells. While abrogating the CpG island-binding activity of FBXL19 or deletion of MED13/13L resulted in reduced CDK8 binding at these sites, it did not lead to an appreciable effect on gene expression in the ES cell state (*Figure 6* and *Figure 6—figure supplement 1C,D*). This suggests that CDK-Mediator is not required for the repression of these genes in ES cells. Instead, these genes fail to properly activate during differentiation (*Figure 6*, *Figure 6—figure supplement 1*, and *Figure 7D*). These observations are in line with several reports that support the idea that CDK-Mediator is also involved in gene activation (*Alarcón et al., 2009*; *Donner et al., 2007*; *Hirst et al., 1999*; *Donner et al., 2010*; *Galbraith et al., 2013*). Therefore, in the context of mouse ES cells, we propose that CDK-Mediator may be required to prime silent developmental genes for future gene activation through a mechanism that relies on the recognition of CpG island promoters by FBXL19. This priming could contribute to gene activation during differentiation through one of the several mechanisms by which CDK-Mediator has been proposed to affect gene transcription, including regulating RNAPolII pre-initiation complex assembly, polymerase pausing/elongation, or through mediating long range interactions with distal regulatory elements (*Allen and Taatjes, 2015*; *Belakavadi and Fondell, 2010*; *Donner et al., 2010*; *Galbraith et al., 2013*; *Kagey et al., 2010*). Clearly, understanding the defined mechanism by which FBXL19-dependent CDK-Mediator recruitment supports normal gene activation during differentiation remains an important question for future work.

Our finding that FBXL19 can target CDK8 to gene promoters and that this is required to prime the expression of developmental genes during ES cell differentiation is conceptually reminiscent of recent work in yeast and human systems which suggested that CDK8 is required for appropriate re-induction of inducible genes following an initial activation stimulus (*D'Urso et al., 2016*). In this study, CDK8 was found to associate with the promoters of inducible genes following initial activation, even in the absence of appreciable ongoing gene transcription, thereby acting as a form of transcriptional memory (*D'Urso et al., 2016*). Interestingly, this requirement for CDK8 in normal gene activation appears shared in the context of our developmental gene induction paradigm. However, we find that, in the case of developmental genes, previous gene activation may not be required for the recruitment of CDK8. Instead, this could be achieved by FBXL19 directly recognising and targeting CDK8 to CpG islands. Therefore, we prefer to view this as priming as opposed to memory. Nevertheless, collectively these observations point to an important role for CDK-Mediator binding to gene promoters prior to activation and as a way of supporting subsequent gene induction. This provides new evidence that the roles of CDK-Mediator in gene regulation are more complicated than simply conveying activation signals to RNAPolII through recruitment by transcription factors and suggests the complex may have evolved to play a unique role in regulating gene expression in stem cells and during development.

## Materials and methods

### Cell culture

Mouse ES cells were cultured on gelatine-coated dishes in DMEM (Thermo Fisher scientific) supplemented with 15% fetal bovine serum (BioSera), L-Glutamine, beta-mercaptoethanol, non-essential amino acids, penicillin/streptomycin (Thermo Fisher scientific) and 10 ng/mL leukemia-inhibitory factor. *Fbxl19*^fl/fl^ ES cells were treated with 800 nM 4-hydroxytamoxifen (Sigma) for 96 hr in order to delete the ZF-CxxC domain. For RA differentiation of ES cells, $2.5 \times 10^4$ cells/cm$^2$ were allowed to attach to gelatinised dishes (~12 hr) and treated with 1 µM retinoic acid (Sigma-Aldrich) in EC-10 medium (DMEM supplemented with 10% fetal bovine serum, L-Glutamine, beta-mercaptoethanol, non-essential amino acids and penicillin/streptomycin) for 72 hr. For embryonic body differentiation, $2 \times 10^6$ cells were plated on non-adhesive 10 cm dishes in EC-10 medium and cultured for the indicated days. For generation of stable cell lines, E14 ES cells were transfected using Lipofectamine 2000 (Thermo Fisher scientific) following manufacturer's instructions. Stably transfected cells were selected for 10 days using 1 µg/ml puromycin and individual clones were isolated and expanded in the presence of puromycin to maintain the transgene expression. TOT2N E14 cells used for TetR targeting experiments were previously described (*Blackledge et al., 2014*). 293 T cells were cultured in EC-10 media. Transient overexpression of FBXL19 was performed by transfecting 293 T cells using Lipofectamine 2000 (Thermo Fisher scientific) followed by selection with 400 ng/µl G418 for 48 hr.

Proteasome inhibition was performed for 4 hr using 10 µM MG132 inhibitor (Sigma-Aldrich). All cell lines generated and grown in the Klose and Koseki labs were routinely tested for mycoplasma infection.

## Generation of *Fbxl19*$^{\Delta CXXC}$ conditional knock-out mouse

The targeting construct was generated from C57BL6/J mouse genomic sequence spanning mm9 chromosome 7: 134,888,487–134,895,358 containing the exons 1 to 6 of *Fbxl19* genomic region. Recombination was carried out by Gateway system (Life Technologies). One of the loxP sequences was inserted at mm9 chr7:134,891,537, and FRT flanked PGK-neo was inserted at mm9 chr7:134,892,000 together with an additional loxP sequence. The targeting construct was electroporated into M1 ES cells to obtain targeted insertion. Clones of targeted ES cells were aggregated with eight-cell embryos to generate the targeted mouse line. The *Fbxl19*$^{fl/fl}$ line was generated by removal of the PGK-neo marker gene by mating the targeted mice with mice expressing FLP recombinase. These *Fbxl19*$^{fl/fl}$ mice were further mated with mice harboring the *ROSA26-CreErt2* locus to generate *Fbxl19*$^{fl/fl}$:*ROSA26-CreErt2*$^{+/-}$ mice, from which the *Fbxl19*$^{fl/fl}$ ES cells used in this study were derived.

## Generation of *Med13/13l*$^{fl/fl}$ conditional knock-out ES cells line

In order to generate a conditional *Med13/13l*$^{fl/fl}$ ES cell line, we inserted loxP sites downstream of exons 7 and upstream of exons 8 of the *Med13* and *Med13l* genes. The targeting constructs for the insertion of each loxP site were designed to have 150 bp homology arms flanking the loxP site and to carry a mutated PAM sequence to prevent retargeting by the Cas9 enzyme. The targeting constructs were purchased from GeneArt Gene Synthesis (Thermo Fisher scientific). The pSpCas9(BB)—2A-Puro(PX459)-V2.0 vector was obtained by Addgene (#62988). sgRNAs were designed to specifically target the desired genomic region for each loxP insertion (http://crispor.tefor.net/crispor.py) and were cloned into the Cas9 vector as previously described (*Ran et al., 2013*). ES cells that express the ERT2-Cre recombinase from the *ROSA26* locus (*ROSA26:ERT2-Cre*) were used. First, the downstream loxP sites for *Med13* and *Med13l* were targeted. *ROSA26:ERT2-Cre* ES cells were transiently co-transfected with 1 µg of each Cas9-sgRNA plasmid and 3.5 µg of each targeting construct using Lipofectamine 3000 (Thermo Fischer Scientific). Successfully transfected ES cells were selected for 48 hr with 1 µg/ml puromycin. Individual clones were screened by genotyping PCR to identify correctly targeted homozygous clones. A clone homozygous for both *Med13* and *Med13l* LoxP1 sites was then used to target the upstream loxP sites using the same transfection protocol and screening strategy. Correct loxP targeting was verified by sequencing of the genomic region surround the loxP sites.

## DNA constructs

For generation of FBXL19 expression constructs, the full length, ΔCxxC or ΔF-box cDNA of mouse *Fbxl19* (IMAGE ID 6401846, Source Bioscience) was PCR amplified and inserted into a pCAG-IRES-FS2 vector (*Farcas et al., 2012*) via ligation-independent cloning (LIC). Mutation of the ZF-CxxC domain of FBXL19 (K49A) was generated via site-directed mutagenesis using the Quikchange mutagenesis XL kit (Stratagene). To generate TetR-FBXL19 fusion expression plasmid, *Fbxl19* cDNA was cloned into pCAG-FS2-TetR (*Blackledge et al., 2014*) via LIC. For transient overexpression in 293 T cells, full length FBXL19 cDNA was cloned into pcDNA3-2xFlag vector by conventional cloning. All plasmids were sequence-verified by sequencing.

## Nuclear extract preparation and immunoprecipitation

Cells were harvested by scraping in PBS at 4°C, resuspended in 10x pellet volume (PV) of Buffer A (10 mM Hepes pH 7.9, 1.5 mM MgCl$_2$, 10 mM KCl, 0.5 mM DTT, 0.5 mM PMSF, cOmplete protease inhibitor cocktail (Roche)) and incubated for 10 min at 4°C with slight agitation. After centrifugation, the cell pellet was resuspended in 3x PV Buffer A containing 0.1% NP-40 and incubated for 10 min at 4°C with slight agitation. Nuclei were recovered by centrifugation and the soluble nuclear fraction was extracted for 1 hr at 4°C with slight agitation using 1x PV Buffer C (10 mM Hepes pH 7.9, 400 mM NaCl, 1.5 mM MgCl$_2$, 26% glycerol, 0.2 mM EDTA, cOmplete protease inhibitor cocktail). Protein concentration was measured using Bradford assay.

For small-scale co-immunoprecipitation, 600 µg of nuclear extract was diluted in BC150 buffer (50 mM Hepes pH 7.9, 150 mM KCl, 0.5 mM EDTA, 0.5 mM DTT, cOmplete protease inhibitor cocktail). Samples were incubated with the respective antibodies and 25U benzonase nuclease overnight at 4°C. Protein A agarose beads (RepliGen) were blocked for 1 hr at 4°C in Buffer BC150 containing 1% fish skin gelatine (Sigma) and 0.2 mg/ml BSA (NEB). The blocked beads were added to the samples and incubated for 4 hr at 4°C. Washes were performed using BC150 containing 0.02% NP-40. The beads were resuspended in 2x SDS loading buffer and boiled for 5 min to elute the immunoprecipitated complexes.

Purification of FBXL19-FS2 was performed using StrepTactin resin (IBA) as previously described with the exception of the wash buffer used (20 mM Tris pH 8.0, 150 mM NaCl, 0.2% NP-40, 1 mM DTT, 5% glycerol, cOmplete protease inhibitor cocktail) (*Farcas et al., 2012*). Between 10 and 15 mg of nuclear extract was used for each large scale purification. The samples were treated with 75 U/mL benzonase nuclease (Novagen) in order to disrupt nucleic-acid-mediated interactions.

## Mass spectrometry

Samples from FBXL19-FS2 affinity purifications were subjected to in-solution trypsin digestion and mass spectrometry analysis was performed as described previously (*Farcas et al., 2012*). Two biological replicates were performed. A control EV purification was included in each replicate. An interaction with identified proteins was only considered significant if absent from the EV data.

## Antibodies

Antibodies used for IP were rabbit anti-MED12 (A300-774A, Bethyl laboratories), rabbit anti-CDK8 (A302-500A, Bethyl laboratories), rabbit anti-HA (3724, Cell Signaling Technology). A polyclonal antibody against FBXL19 was prepared in-house by rabbit immunisation (PTU/BS Scottish National Blood Transfusion Service) with a recombinant peptide encoding for amino acids 137–336 of mouse FBXL19 protein. The FBXL19 peptide antigen was coupled to Affigel 10 resin (BioRad) and the antibody was affinity-purified and concentrated.

Antibodies used for Western blot analysis were mouse anti-Flag M2 (F1804, Sigma-Aldrich), mouse anti-SKP1 (sc-5281, Santa Cruz), goat anti-CDK8 (sc-1521, Santa Cruz), rabbit anti-MED12 (A300-774A, Bethyl laboratories), rabbit anti-MED13L (A302-420A, Bethyl laboratories), rabbit anti-MED13 (GTX129674, Genetex), rabbit anti-MED1, (A300-793A, Bethyl laboratories), rabbit anti-MED26 (A302-370A, Bethyl laboratories), rabbit anti-T7 (13246, Cell Signaling Technology), rabbit anti-TBP (ab818, Abcam), rabbit anti-RNF20 (11974, Cell Signaling Technology), rabbit anti-ubiquityl-Histone H2B (Lys120) (5546, Cell Signaling Technology), and mouse anti-ubiquityl-Histone H2B (Lys120) (05–1312, Millipore).

## Generation of T7-FBXL19 ES cell line

As the FBXL19 antibody failed to work reliably for Western blot analysis, we generated an *Fbxl19*<sup>fl/fl</sup> ES cell line in which the endogenous *Fbxl19* gene is tagged with a 3xT7-2xStrepII tag by CRISPR/Cas9 knock-in (*Figure 2—figure supplement 1B*). This allowed us to determine the efficiency of the OHT treatment and endogenous IPs. The tag was synthesised by GeneArt (ThermoFischer Scientific) and the targeting construct was generated by PCR amplification to introduce roughly 150 bp homology arms flanking the 3xT7-2xStrepII tag. The PCR product was cloned into pGEM-T Easy Vector (Promega). The pSpCas9(BB)−2A-Puro vector was obtained by Addgene (#48139). A sgRNA was designed to overlap with the stop codon of the *Fbxl19* gene (http://crispr.mit.edu/) and cloned into the Cas9 vector as previously described (*Ran et al., 2013*). *Fbxl19*<sup>fl/fl</sup> ES cells were transiently co-transfected with 1 µg of Cas9-sgRNA plasmid and 3.5 µg of targeting construct using Lipofectamine 3000 (ThermoFischer Scientific). Successfully transfected ES cells were selected for 48 hr with 1 µg/ml puromycin. Individual clones were screened by Western blot and genotyping PCR to identify homozygous targeted clones.

## Chromatin immunoprecipitation

Chromatin immunoprecipitation was performed as described previously (*Farcas et al., 2012*) with slight modifications. Cells were fixed for 45 min with 2 mM DSG (Thermo scientific) in PBS and 12.5 min with 1% formaldehyde (methanol-free, Thermo scientific). Reactions were quenched by the

addition of glycine to a final concentration of 125 µM. After cell lysis and chromatin extraction, chromatin was sonicated using a BioRuptor Pico sonicator (Diagenode), followed by centrifugation at 16,000 × g for 20 min at 4°C. 200 µg chromatin diluted in ChIP dilution buffer (1% Triton-X100, 1 mM EDTA, 20 mM Tris-HCl (pH 8.0), 150 mM NaCl) was used per IP. Chromatin was precleared with protein A Dynabeads blocked with 0.2 mg/ml BSA and 50 µg/ml yeast tRNA and incubated with the respective antibodies overnight at 4°C. Antibody-bound chromatin was purified using blocked protein A Dynabeads for 3 hr at 4°C. ChIP washes were performed as described previously (*Farcas et al., 2012*). ChIP DNA was eluted in ChIP elution buffer (1% SDS, 100 mM NaHCO$_3$) and reversed cross-linked overnight at 65°C with 200 mM NaCl and RNase A (Sigma). The reverse cross-linked samples were treated with 20 µg/ml Proteinase K and purified using ChIP DNA Clean and Concentrator kit (Zymo research).

The antibodies used for ChIP experiments were rabbit anti-FS2 (*Farcas et al., 2012*), rabbit anti-CDK8 (A302-500A, Bethyl laboratories), rabbit anti-MED12 (A300-774A, Bethyl laboratories), mouse anti-MED4 (PRC-MED4-16, DSHB), rabbit anti-SKP1 (12248, Cell Signaling Technology), rabbit anti-KDM2A (*Farcas et al., 2012*), rabbit anti-KDM2B (*Farcas et al., 2012*).

## ChIP-sequencing

All ChIP-seq experiments were carried out in biological duplicates. ChIP-seq libraries for FS2-FBXL19 ChIP were prepared using the NEBNext Fast DNA fragmentation and library preparation kit for Ion Torrent (NEB, E6285S) following manufacturer's instructions. Briefly, 30–40 ng ChIP or input DNA material was used. Libraries were size-selected for 250 bp fragments using 2% E-gel SizeSelect gel (Thermo scientific) and amplified with 6 PCR cycles or used without PCR amplification. Templates were generated with Ion PI Template OT2 200 kit v3 and Ion PI Sequencing 200 kit v3, or with Ion PI IC 200 kit (Thermo scientrific). Libraries were sequenced on the Ion Proton Sequencer using Ion PI chips v2 (Thermo scientific).

ChIP-seq libraries for CDK8 ChIP were prepared using the NEBNext Ultra DNA Library Prep Kit, and sequenced as 40 bp paired-end reads on Illumina NextSeq500 platform using NextSeq 500/550 (75 cycles).

## Reverse transcription and gene expression analysis

Total RNA was isolated using TRIzol reagent (Thermo scientific) following manufacturer's instructions and cDNA was synthesized from 400 ng RNA using random primers and ImProm-II Reverse Transcription system kit (Promega). RT-qPCR was performed using SensiMix SYBR mix (Bioline). Idh1 and Atp6IP1 genes were used as house-keeping controls.

## 4sU-labeling of nascent RNA transcripts

For labeling of nascent RNA transcripts, cells were grown in 15 cm culture dishes. Labeling was performed for 20 min at 37°C by the addition of 50 mM 4-thiouridine (T4509, Sigma) to the culture medium. After the incubation, the medium was removed and RNA was isolated using TRIzol reagent (Thermo scientific) following manufacturer's instructions. The total RNA was treated with TURBO DNA-free kit (Ambion, Thermo scientific) in order to remove contaminating genomic DNA. 300 µg RNA was biotinylated using 600 µg Biotin-HPDP (21341, Pierce, Thermo scientific) in Biotinylation buffer (100 mM Tris-HCl pH 7.4, 10 mM EDTA). The reaction was carried out for one and half hour on a rotor at room temperature. Unincorporated Biotin-HPDP was removed by chloroform/isoamylalcohol (24:1, Sigma) wash followed by isopropanol precipitation of the biotinylated RNA. Labeled biotinylated RNA was isolated using µMacs Streptavidin Kit (130-074-101, Miltenyi) following manufacturer's instructions and purified using RNeasy MinElute Cleanup kit (Qiagen). The quality of the RNA was confirmed using the RNA pico Bioanalyser kit (Agilent).

## 4sU RNA sequencing

Up to 1 µg purified 4sU-labelled RNA was used to prepare libraries for 4sU-RNA-seq. Ribosomal RNA was removed using NEBNext rRNA Depletion Kit and libraries were prepared using NEBNext Ultra Directional RNA Library Prep Kit for Illumina (NEB) following manufacturer's instructions. Library quality was assessed using the High-sensitivity DS DNA Bioanalyser kit (Agilent) and the concentration was measured by quantitative PCR using KAPA Library quantification standards for

Illumina (KAPA Biosystems). All 4sU RNA-seq experiments were carried out in biological triplicates. 4sU RNA-seq libraries were sequenced as 80 paired-end reads on Illumina NextSeq500 platform using NextSeq 500/550 High Output Kit v2 (150 cycles).

### Analysis of high-throughput sequencing

Sequencing reads were aligned to the mouse genome (mm10) using bowtie2 (*Langmead and Salzberg, 2012*) with the '–no-mixed' and '–no-discordant' options. Reads that mapped more than once to the genome were discarded. For 4sU RNA-seq analysis, rRNA reads were initially removed by aligning the data to mouse rRNA genomic sequences (GenBank: BK000964.3) using bowtie2. The rRNA-depleted reads were next aligned against mm10 genome using the STAR RNA-seq aligner (*Dobin et al., 2013*). To improve mapping of nascent, intronic sequences, a second alignment step with bowtie2 was included using the reads which failed to map using STAR. PCR duplicates were removed using samtools (*Li et al., 2009*). Biological replicates were randomly downsampled to the same number of reads for each individual replicate and merged for visualisation. Bigwig files were generated using MACS2 (*Zhang et al., 2008*) and visualised using the using the UCSC Genome Browser (*Raney et al., 2014*).

Peak calling was performed using the MACS2 function with the '-broad' option using the biological replicates with matched input (*Zhang et al., 2008*). Peaks mapping to a custom 'blacklist' of artificially high genomic regions were discarded. Only peaks called in both replicates were considered. Differential analysis of CDK8 binding was done using DiffReps (*Shen et al., 2013*) and the called differential regions were overlapped with CDK8 peaks. Changes in binding of log2FC less than −0.5 or more than 0.5 with adjusted p-value below 0.05 were considered significant.

Differential expression analysis was performed using the DESeq2 package (*Love et al., 2014*), version 1.6.3, with R version 3.1.1. Counts were quantified using the summarizeOverlaps() function in R in the mode 'Union' and inter.feature = FALSE. Genes with an adjusted p-value of below 0.1 and a fold change of at least 1.5 were considered differentially expressed. Statistical analysis was performed using two-sample Wilcoxon rank sum test. Gene ontology analysis was done using DAVID 6.8 (*Huang et al., 2009*). For CDK8 differentially bound peaks, all CDK8 peaks were provided as background. For misregulated genes analysis, all RefSeq genes were provided as background. ATAC peaks were obtained from *King and Klose (2017)*.

### Accession numbers

ChIP-seq and RNA-seq data from the present study are available to download at GSE98756. Previously published studies used for analysis include mouse ES cell BioCAP (GSE43512, [*Long et al., 2013b*]), Kdm2B ChIP-seq (GSE55698, [*Blackledge et al., 2014*]), Kdm2A ChIP-seq (GSE41267, [*Farcas et al., 2012*]), H3K27me3 ChIP-seq (GSE83135, [*Rose et al., 2016*]), H3K4me3 ChIP-seq (GSE93538, [*Brown et al., 2017*]).

## Acknowledgements

We thank Anne Turberfield for help with the 4sU-RNAseq protocol and useful discussion, and Hamish King for assistance with computational analysis and critical comments and suggestions. We thank David Brown for initiating the FBXL19 experiments, and Anne Turberfield and Neil Blackledge for critical reading of the manuscript. We thank Ed Hookway and Udo Oppermann for sequencing support on the NextSeq500 at the Botnar sequencing facility in Oxford. Work in the Klose laboratory is supported by the Wellcome Trust, the Lister Institute of Preventive Medicine and the European Research Council. TK and HK are supported by the AMED-CREST programme from the Japan Agency for Medical Research and Development. AF is supported by a Sir Henry Wellcome Postdoctoral Fellowship.

## Additional information

### Funding

| Funder | Grant reference number | Author |
|---|---|---|
| Sir Henry Wellcome Postdoctoral Fellowship | 110286/Z/15/Z | Angelika Feldmann |
| Japan Agency for Medical Research and Development | | Haruhiko Koseki |
| Wellcome | 098024/Z/11/Z | Robert J Klose |
| European Research Council | 681440 | Robert J Klose |
| Lister Institute of Preventive Medicine | | Robert J Klose |

The funders had no role in study design, data collection and interpretation, or the decision to submit the work for publication.

### Author contributions

Emilia Dimitrova, Conceptualization, Data curation, Formal analysis, Validation, Investigation, Visualization, Methodology, Writing—original draft, Writing—review and editing; Takashi Kondo, Resources, Data curation, Formal analysis, Methodology; Angelika Feldmann, Resources, Software, Formal analysis; Manabu Nakayama, Yoko Koseki, Resources, Contributed to transgenic mouse generation and embryonic stem cell derivation and characterisation; Rebecca Konietzny, Benedikt M Kessler, Resources, Contributed to mass spectrometry sample processing and analysis; Haruhiko Koseki, Supervision, Funding acquisition, Investigation, Project administration; Robert J Klose, Conceptualization, Supervision, Funding acquisition, Investigation, Writing—original draft, Project administration, Writing—review and editing

### Author ORCIDs

Emilia Dimitrova (iD) http://orcid.org/0000-0001-5669-1240
Haruhiko Koseki (iD) http://orcid.org/0000-0001-8424-5854
Robert J Klose (iD) http://orcid.org/0000-0002-8726-7888

### Decision letter and Author response

Decision letter https://doi.org/10.7554/eLife.37084.036
Author response https://doi.org/10.7554/eLife.37084.037

## Additional files

### Supplementary files
• Transparent reporting form
DOI: https://doi.org/10.7554/eLife.37084.019

### Data availability

Sequencing data generated in this study have been deposited in GEO under accession code GSE98756.

The following dataset was generated:

| Author(s) | Year | Dataset title | Dataset URL | Database, license, and accessibility information |
|---|---|---|---|---|
| Dimitrova E | 2017 | FBXL19 recruits CDK8-Mediator to CpG islands and primes developmental genes for activation during lineage commitment | https://www.ncbi.nlm.nih.gov/geo/query/acc.cgi?acc=GSE98756 | Publicly available at the NCBI Gene Expression Omnibus (accession no. GSE98756) |

The following previously published datasets were used:

| Author(s) | Year | Dataset title | Dataset URL | Database, license, and accessibility information |
|---|---|---|---|---|
| Blackledge NP, Farcas AM, Kondo T, King HW, McGouran JF, Hanssen LL, Ito S, Cooper S, Kondo K, Koseki Y, Ishikura T, Long HK, Sheahan TW, Brockdorff N, Kessler BM, Koseki H, Klose RJ | 2014 | Variant PRC1 complex dependent H2A ubiquitylation drives PRC2 recruitment and polycomb domain formation | https://www.ncbi.nlm.nih.gov/geo/query/acc.cgi?acc=GSE55698 | Publicly available at the NCBI Gene Expression Omnibus (accession no. GSE55698) |
| Farcas AM, Blackledge NP, Sudbery I, Long HK, McGouran JF, Rose NR, Lee S, Sims D, Cerase A, Sheahan TW, Koseki H, Brockdorff N, Ponting CP, Kessler BM, Klose RJ | 2012 | KDM2B links the Polycomb Repressive Complex 1 (PRC1) to recognition of CpG islands | https://www.ncbi.nlm.nih.gov/geo/query/acc.cgi?acc=GSE41267 | Publicly available at the NCBI Gene Expression Omnibus (accession no. GSE41267) |
| Long HK, Sims D, Heger A, Blackledge NP, Kutter C, Wright ML, Grützner F, Odom DT, Patient R, Ponting CP, Klose RJ | 2013 | Epigenetic conservation at gene regulatory elements revealed by non-methylated DNA profiling in seven vertebrates | https://www.ncbi.nlm.nih.gov/geo/query/acc.cgi?acc=GSE43512 | Publicly available at the NCBI Gene Expression Omnibus (accession no. GSE43512) |
| Rose NR, King HW, Blackledge NP, Fursova NA, Ember KJ, Fischer R, Kessler BM, Klose RJ | 2016 | RBYP stimulates PRC1 to shape chromatin-based communication between polycomb repressive complexes | https://www.ncbi.nlm.nih.gov/geo/query/acc.cgi?acc=GSE83135 | Publicly available at the NCBI Gene Expression Omnibus (accession no. GSE83135) |
| Brown DA, Di Cerbo V, Feldmann A, Ahn J, Ito S, Blackledge NP, Nakayama M, McClellan M, Dimitrova E, Turberfield AH, Long HK, King HW, Kriaucionis S, Schermelleh L, Kutateladze TG, Koseki H, Klose RJ | 2017 | CFP1 engages in multivalent interactions with CpG island chromatin to recruit the SET1 complex and regulate gene expression. | https://www.ncbi.nlm.nih.gov/geo/query/acc.cgi?acc=GSE93538 | Publicly available at the NCBI Gene Expression Omnibus (accession no. GSE93538) |

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
