## [Decision Letter]

[Editors’ note: a previous version of this study was rejected after peer review, but the authors submitted for reconsideration. The first decision letter after peer review is shown below.]

Thank you for submitting your work entitled "FBXL19 recruits CDK8-Mediator to CpG islands and primes developmental genes for activation during lineage commitment" for consideration by *eLife*. Your article has been reviewed by four peer reviewers, one of whom is a member of our Board of Reviewing Editor and the evaluation has been overseen by a Senior Editor.

The reviewers have discussed the reviews with one another and the Reviewing Editor has drafted this letter to crystallize our concerns going forward. The work presented was deemed important and interesting but key issues reduced enthusiasm for the manuscript considerably. Given that these revisions are substantial and likely to take a significant amount of time, we are currently declining the present version of the paper. That said, if you feel you can successfully address all the concerns raised, we will be open to consider a revised submission, which would be treated as a de novo submission but sent to the same editors and, if possible, referees for re-assessment.

Summary:

The reviewers considered the manuscript to be strong, reporting a thorough characterization of FBXL19, the least-well studied of three proteins that contain a ZF-CxxC domain. The authors demonstrate that over-expressed FBXL19 accumulates in the nucleus, and that a majority of its genome-wide binding is at sites associated with CpG islands. Affinity purification of FS2-tagged FBXL19 indicated a physical association with CDK8-containing Mediator complex, and a TetR-FBXL19 fusion protein was able to recruit CDK8 to a chromosomally-integrated artificial TetO array, suggesting that FBXL19 can recruit CDK8 to chromatin. Conditional deletion of the ZF-CxxC domain of FBXL19 led to reduction of CDK8 binding at CpG islands in ES cells. Many of these sites are associated with silent developmental genes that are no longer activated during ES cell differentiation in the FBXL19 mutants. Finally, deletion of the ZF-CxxC domain was shown to be embryonic lethal in mice. Together, these findings suggest that FBXL19 may prime developmental genes for subsequent activation during differentiation, possibly involving recruitment of CDK8. Overall, this represents an interesting finding and potentially a novel role for CDK8. However, concerns about the degree to which this represents a significant advance over previous FBXL19 studies, some missing controls, and a lack of a direct test of CDK8 requirement for expression of these genes resulted in lessened enthusiasm for this manuscript.

The reviewers discussed these concerns and summarized their critiques into the following points.

Essential revisions:

1) A key conclusion of the manuscript is that CDK8-Mediator is required for induction of developmental genes regulated by FBXL19, but this is actually not proven in the manuscript. The authors should deplete CDK8 (and/or CDK19, see comment #2 below) and test for the impact on induction of FBXL19-dependent genes.

2) Throughout the paper, the authors refer to CDK8-Mediator. Given that there are two paralog kinases associated with the Mediator complex, CDK8 and CDK19, the authors should test for the presence of CDK19 and clarify whether the interaction is exclusive to CDK8-containing complexes or not. In Figure 2A, are the CDK8 peptides identified unique to CDK8 or could they map to CDK19 as well? In Figure 2C, please add western blot for CDK19. As for point #1 above, is CDK19 required for control of FBXL19-regulated genes? Additionally, the authors should demonstrate that in the "EV" control lane, at the similar molecular weight, these candidate hits didn't show up in the cut gel, or at least with significantly less numbers of peptides.

3) If the ZF-CxxC mediates the interaction between FBXL19 and chromatin, then which functional domain(s) on FBXL19 bind to CDK8-Mediator complex? Lee et al., claimed that the F-box is critical for the interaction between FBXL19 and RNF20. Does the same protein structure bind to CDK8?

4) Most of the findings are based on overexpression of FBXL19 – what is the level relative to endogenous FBXL19? High-level over-expression could drive nuclear accumulation and other observations (including Figure 3). Authors should provide data demonstrating the degree of overexpression produced in the key experiments.

5) FBXL19 binding to CpG islands and its requirement in ES differentiation has been described previously: (https://www.ncbi.nlm.nih.gov/pubmed/28453857) "Our genome-wide target mapping unveils the preferential occupancy of Fbxl19 on CpG island-containing promoters […]" "Moreover, we reveal that Fbxl19 is critical for proper differentiation of ES cells […]" With respect to this, the authors should clarify the novelty of their results presented in the current manuscript.

6) The conclusion that FBXL19 does not appear to regulate CDK8-Mediator levels is not fully supported by the data. Figure 2—figure supplement 1H needs to be repeated so that CDK8 bands are not over-exposed as there does appear to be a change in amount/mobility when overexpressed -/+ MG132. Furthermore, there seems to be a noticeable increase of MED12 protein upon FBXL19 overexpression, which was even more obvious when cells were treated with MG132.

7) In Figure 4, CDK8 recruitment has only been tested for the FBXL19 ZF-CxxC deletion -is this recapitulated for depletion of the whole protein? Given that Figure 4B appears to show a decrease in the levels of the deletion protein this is an important point to verify.

8) The authors need to show that OHT treatment of the delCxxC ES line leads to reduced binding of FBXL19 in conditions used for CDK8 ChIPseq -several genes would be ok -these same genes should be used to validate CDK8 CHIPseq signal.

9) Supplemental tables with differential gene expression results should be provided -or at least working gene lists used to reach the conclusions in the manuscript. Full GO enrichment results should be included as supplementary tables so that specific gene lists can be examined.

10) Various figure panels are missing statistical tests: Figure 4C needs statistical tests, e.g. KS/Wilcoxon -See Figure 4E; Figure 4G needs statistical tests; Figure 4H and 4I need boxplots and need statistical tests eg KS/Wilcoxon as for E/F; Figure 4H-J should include FBXL19-bound groups; Figure 6B,E,F need statistical tests.

11) It is very nice that the authors showed that genome-wide chromatin binding of FBXL19 was completely lost when the ZF-CxxC domain was deleted or mutated. To further confirm that it is indeed this ZF-CxxC domain mediating interaction with the non-methylated CpG islands, some controls are necessary. For example, in Figure 1—figure supplement 1D, the authors showed that almost all of the FBXL19 binding sites are overlapped with NMIs locations, with only 93 exceptions. Then are the signals at these 93 sites also changed in the FBXL19 mutant ChIPSeq data? When the cell DNA methylation levels are modulated, will the binding pattern of FBXL19 changed accordingly?

[Editors’ note: what now follows is the decision letter after the authors submitted for further consideration.]

Thank you for submitting your article "FBXL19 recruits CDK-Mediator to CpG islands of developmental genes priming them for activation during lineage commitment" for consideration by *eLife*. Your article has been reviewed by three peer reviewers, one of whom is a member of our Board of Reviewing Editors and the evaluation has been overseen by James Manley as the Senior Editor. The reviewers have opted to remain anonymous.

The reviewers have discussed the reviews with one another and the Reviewing Editor has drafted this decision to help you prepare a revised submission.

The revised manuscript by Dimitrova et al., is an excellent report describing the characterization of FBXL19, a member of the ZF-CxxC domain-containing family of proteins, and its role in recruiting CDK-Mediator to developmental genes in stem cells.

As for other members of this family, the authors show that FBXL19 recognizes unmethylated CpGs in a ZF-CxxC-dependent manner. Unlike the lysine demethylases KDM2A and KDM2B in this protein family, FBXL19 lacks a JMJ-C domain. Instead, the authors report that FBXL19 interacts with the CDK-Mediator complex. The authors then embark on an experimental tour-de-force to demonstrate that FBXL19 is required for recruitment of CDK-Mediator to a subset of genes containing CpG islands in ES cells, mostly genes involved in lineage commitment. Interestingly, these genes are transcribed at low levels in ES cells when bound by FBXL19 and CDK-Mediator, but their induction is compromised in the absence of FBXL19. This indicates that FBXL19-CDK8-Mediator is required to 'prime' these genes in ES cells and their subsequent induction upon differentiation stimuli. Finally, the authors demonstrate that FBXL19 depletion is embryonic lethal. Overall, the manuscript is very well written, and the data is of excellent quality.

This is a new submission of a paper that was originally reviewed by the same referees and found to be lacking data to support some of the key conclusions. The Senior Editor opened up the possibility of a new submission without a precise timeline and the authors have taken the opportunity to address the major concerns raised in the original review. The referees are now supportive of publication of a revised manuscript addressing the following issues:

Essential revisions:

1) In AP-MS (Figure 2A), what is the background value of emPAI? Are there any other FBXL19-interacting proteins that show significantly high levels of emPAI in the list? How significant are the components of CDK-Mediator complex represented in all the hits?

2) In Figure 3, the authors demonstrated that FBXL19 was able to recruit CDK-Mediator complex, such as CDK8 and MED12, to artificially designed chromatin regions. What about CXXC-domain or F-box deletion mutants in this assay?

3) In Figure 4B, it is not very clear which band(s) are exactly indicated as full-length FBXL19 and which one(s) are CXXC deletion? Why are there double bands? Did the author use T7 antibody to detect the endogenously introduced protein?

4) In Figure 4C, what proportions of each group of CDK8 binding sites overlap with FBXL19? In theory, the author expects to see that CDK8 sites showing binding intensity changes upon tamoxifen treatment would tend to overlap more with FBXL19.

5) In Figure 4—figure supplement 1C, please use a Venn Diagram to show directly the number of peaks of CDK8, FBXI19 and NMI overlapping with each other. This would be more straightforward.

6 In Figure 5, please perform the same GSEA or functional annotation analysis on genes that are associated with unchanged CDK8 binding sites upon FBXL19 CXXC-domain deletion.

7) In Figure 6B, what is the result for genes that are associated with increased CDK8 binding signals in FBXL19 CXXC-domain deletion mutant upon RA-induced differentiation?

8) In Figure 6C, even though the authors claimed that those upregulated genes (889) in FBXL19 CXXC-domain deletion mutant upon RA treatment were not involved in developmental processes, it is still interesting to find that significantly more genes were activated upon CXXC-domain deletion. What explanation can the authors provide for this finding?

9) In Figure 7, the authors claim that FBXL19 relies on CDK-mediator complex to activate the specific set of developmental genes. Is it possible to test this hypothesis in the FBXL19 CXXC-deletion system they established in Figure 6? In wild-type ESC, depletion of MED13 will have similar effects as FBXL19 CXXC deletion, whereas in FBXL19 mutant, the effects would be much alleviated?

10) In Figure 8, why did heterozygous deletion of FBXL19 CXXC-domain show even better developmental status than the wild type?

11) The authors demonstrated that the CXXC-domain mediates the binding of FBXL19 at NMIs, where F-Box mediates its interaction with the CDK-Mediator complex. What about CDK8 chromatin binding or gene expression patterns upon F-Box deletion of FBXL19? Will the authors see the same results as what they found in CXXC-domain deletion mutant?

---

## [Author Response]

[Editors’ note: the author responses to the first round of peer review follow.]

Essential revisions:1) A key conclusion of the manuscript is that CDK8-Mediator is required for induction of developmental genes regulated by FBXL19, but this is actually not proven in the manuscript. The authors should deplete CDK8 (and/or CDK19, see comment #2 below) and test for the impact on induction of FBXL19-dependent genes.

We agree with the reviewers that testing the impact that removal of CDK-Mediator has on the induction of FBXL19 target genes is an important experiment. To disrupt CDK-Mediator we have developed a new cell system to remove MED13 and its closely related paralogue MED13L (both of which are expressed in mouse ES cells) in an inducible manner. We chose to delete both MED13 and MED13L because they are thought to play an interchangeable role in physically linking the CDK subunits to the core Mediator and to be required to form an intact CDK-Mediator complex (Kuesel et al., 2009; Tsai et al., 2013). To achieve this, we exploited gene targeting to insert loxP sites around essential exons in both MED13 and MED13L. We also engineered these cells to express a tamoxifen inducible form of Cre recombinase allowing inducible deletion of MED13/13L (Figure 7—figure supplement 1). In new Figure 7, we demonstrate that the addition of tamoxifen leads to complete loss of MED13/13L protein. Furthermore, we show by ChIP that CDK8 no longer occupies its target sites. We then examined FBXL19 target genes after loss of CDK-Mediator binding and found that they are no longer appropriately induced during differentiation (Figure 7D), in keeping with a requirement for CDK-Mediator in FBXL19dependent gene activation. We thank the reviewers for suggesting this important experiment as it now robustly supports and strengthens our key argument that FBXL19 recruits the CDK-Mediator complex to the CpG islands of inactive developmental genes to prime them for appropriate activation during differentiation. We have now edited the text throughout the manuscript to reflect these important new observations.

2)Throughout the paper, the authors refer to CDK8-Mediator. Given that there are two paralog kinases associated with the Mediator complex, CDK8 and CDK19, the authors should test for the presence of CDK19 and clarify whether the interaction is exclusive to CDK8-containing complexes or not. In Figure 2A, are the CDK8 peptides identified unique to CDK8 or could they map to CDK19 as well? In Figure 2C, please add western blot for CDK19. As for point #1 above, is CDK19 required for control of FBXL19-regulated genes?

As indicated by the reviewer, CDK8 has a highly similar paralogue, CDK19. As suggested, we have now examined our mass spectrometry results in more detail to determine if we find any peptides that also map to CDK19. From this analysis, we have identified five different peptides covering CDK8/CDK19. Four are common to both CDK8 and CDK19, and one is unique for CDK8 (Author response image 1). In our original submission, we had referred to an interaction between FBXL19 and the CDK8-Mediator complex given the identification of a peptide unique to CDK8. However, we agree that this was an oversight as one cannot exclude the possibility that FBXL19 also associates with CDK19-containing Mediator complexes. In fact, given that CDK19 is also expressed in mouse ES cells and is thought to function interchangeably with CDK8, it seems likely that FBXL19 will also interact with CDK19Mediator. Our CKD8 antibody uniquely recognises CDK8, but we have been unable to source an antibody that specifically recognises CDK19, limiting our ability to test whether CDK19 also associates with FBXL19. An antibody with specificity to CDK19 (Atlas antibody HPA007053) has previously been published, but this is currently unavailable, and the company informed us that this is due to a loss of specificity in new batches. Given that we cannot exclude the possibility that FBXL19 also associates with Mediator complexes containing CDK19, we have now added a sentence describing this possibly in the text as follows (subsection “FBXL19 interacts with the CDK-Mediator complex in ES cells”):

*‘*CDK8 and its paralogue CDK19 share 77% amino acid identity (89% similarity) (Audetat et al., 2017) and four out of the five peptides identified by mass spectrometry were common between the two proteins (data not shown). Therefore, it is likely that FBXL19 is able to interact with both CDK8- and CDK19- containing Mediator complexes.’

Furthermore, because the FBXL19-associated Mediator complex could contain either CDK8 or CDK19, we have edited the text throughout the manuscript to replace ‘CDK8-Mediator’ with ‘CDK-Mediator’. In future work it will be interesting to understand in more detail if CDK8- and CDK19-containing Mediator complexes have redundant or unique roles in gene activation during differentiation.

**Author response image 1. respfig1:** CDK8/19 peptide sequences identified by mass spectrometry. A CDK8 unique peptide is indicated in red. The other four peptides, indicated in yellow, are shared between CDK8 and CDK19.

Additionally, the authors should demonstrate that in the "EV" control lane, at the similar molecular weight, these candidate hits didn't show up in the cut gel, or at least with significantly less numbers of peptides.

We carried out mass spectrometry analysis on affinity-purified FBXL19 by in-solution digestion followed by LC-MS. This was to overcome the limitations inherent to cutting bands from SDS-PAGE gels and the isolation of tryptic peptides from gel slices. Importantly, we only considered a protein to be an FBXL19 interactor in our mass spectrometry analysis if it was absent from matched EV control purifications analysed using the same in-solution approach. It would have been beneficial to have described this approach in more detail in the initial submission, so we have now expanded our description of these experiments in the revised Material and Methods. Furthermore, we validated that CDK-Mediator components were not found in the ‘EV’ control purifications by Western blot analysis (Figure 2C).

3) If the ZF-CxxC mediates the interaction between FBXL19 and chromatin, then which functional domain(s) on FBXL19 bind to CDK8-Mediator complex? Lee et al. claimed that the F-box is critical for the interaction between FBXL19 and RNF20. Does the same protein structure bind to CDK8?

We thank the reviewers for this suggestion and agree this is an important point. To address this, we performed transient overexpression of FBXL19 followed by co-immunoprecipitation comparing wildtype FBXL19 with versions of FBXL19 in which individual domains had been removed. We found that deletion of the F-box largely disrupted the interaction with CDK-Mediator, while removing the ZF-CxxC domain had little effect on this interaction. We have included these new observations in the revised manuscript (subsection “FBXL19 interacts with the CDK-Mediator complex in ES cells”, Figure 2—figure supplement 1E). We also attempted to determine whether the LRR domain of FBXL19 was required for this interaction but unfortunately removing this portion of the protein seemed to render it unstable (unpublished observations) limiting our capacity to draw any meaningful interpretation.

4) Most of the findings are based on overexpression of FBXL19 – what is the level relative to endogenous FBXL19? High-level over-expression could drive nuclear accumulation and other observations (including Figure 3). Authors should provide data demonstrating the degree of overexpression produced in the key experiments.

We have examined FBXL19 transcript levels in the transgene lines and they are roughly eight times that of wild type (Author response image 2). Despite exhaustive efforts we have been unable to generate an antibody that recognizes the endogenous protein in cellular extracts by Western blot. Therefore, we have been unable to directly compare endogenous FBXL19 protein levels to the FBXL19 levels in transgene containing lines. Nevertheless, we are confident in the conclusions from our key experiments. We have validated that endogenous FBXL19 is in the nucleus and interacts with CDKMediator. This was demonstrated by immunoprecipitating endogenous FBXL19 protein from nuclear extracts and using Western blot to show that FBXL19 associates with CDK8 and MED12 (Figure 2—figure supplement 1D), consistent with our transgene experiments (Figure 2).

Given that we know FBXL19 is in the nucleus and interacts with CDK-Mediator, tethering experiments in Figure 3 were used to test if de novo targeting of FBLX19 to naïve chromatin would result in recruitment of CDK-Mediator. This showed that FBXL19 binding was sufficient to recruit CDK-Mediator (Figure 3B), in agreement with our observations that when the capacity of FBXL19 to bind CpG island chromatin is disrupted, CDK8 is no longer appropriately recruited to a subset of CpG island-associated target genes (Figure 4).

**Author response image 2. respfig2:** Fbxl19 overexpression levels. RT-PCR showing the expression levels of FS2-Fbxl19 relative to endogenous Fbxl19 in the empty vector (EV) control line. Data is the average of three biological replicates. Error bars represent standard deviation.

5) FBXL19 binding to CpG islands and its requirement in ES differentiation has been described previously: (https://www.ncbi.nlm.nih.gov/pubmed/28453857) "Our genome-wide target mapping unveils the preferential occupancy of Fbxl19 on CpG island-containing promoters […]" "Moreover, we reveal that Fbxl19 is critical for proper differentiation of ES cells […]" With respect to this, the authors should clarify the novelty of their results presented in the current manuscript.

During the preparation of our manuscript, the manuscript cited by the reviewer was published in NAR. It reported that FBXL19 occupied CpG islands and appeared to play a role in cellular differentiation. However, we would like to point out that, although our observations were similar with respect to these two points, the main discoveries resulting from our careful and systematic study are completely distinct from that of the published study. In direct contrast to the published work, we find no evidence for any connection between FBXL19 and RNF20 using unbiased biochemistry and mass spectrometry (Figure 2 and Figure 2—figure supplement 1F). Furthermore, using our conditional genetic ablation strategies we do not observe any effect of FBXL19 loss on H2BK120ub1 (Figure 2—figure supplement 1G). Therefore, we are unable to reproduce the central findings of the published study. In our opinion this severely limits and brings into question the validity and, therefore, the novelty of the published work. Our observations likely differ from those of the NAR study because we use unbiased as opposed to candidate-based (guess work) biochemistry and conditional knockout strategies as opposed to stable knockdown cell lines that are prone to indirect and secondary effects due to the long-term culture necessary for their generation. Nevertheless, we agree with the reviewer that the wording of the indicated sentences could have been more inclusive and contrasting of the published work and have clarified this in the revised manuscript (Introduction, Results section, subsection “FBXL19 interacts with the CDK-Mediator complex in ES cells” and subsection “Removing the CpG island-binding domain of FBXL19 results in a failure to induce developmental genes during ES cell differentiation”).

We would also like to take this opportunity to clarify the novelty of our study and the importance of its central conclusions. (1) By using unbiased biochemical approaches, we discover a link between FBXL19 and CDK-Mediator in mouse ES cells. (2) Using conditional genetic ablation, de novo targeting strategies, and genomics, we go on to show that FBLX19 can recruit CDK-Mediator to chromatin and the CpG island promoters of silent developmental genes in ES cells. (3) In the absence of FBXL19, these genes are no longer appropriately activated upon differentiation, providing an explanation for why normal differentiation cannot be achieved in the absence of FBXL19. (4) In the revised study, based on the reviewer’s suggestions (point 1), we now demonstrate that FBXL19 target genes also rely on CDKMediator for their activation, validating our biochemistry and genomics. (5) Finally, we discover that loss of FBXL19 leads to abnormal development and early embryonic lethality, in agreement with the inability of cells to appropriately regulate gene expression during development.

Therefore, our discoveries are novel in describing a completely new modality by which FBXL19 and CpG islands prime genes for future expression. Furthermore, we provide compelling new evidence that CDKMediator can be recruited to gene promoters independently of classical transcription factors and transcription itself to prime CpG island-associated genes for activation. These new discoveries force us, and the field, to think differently about how CpG islands integrate and regulate gene expression, while also provide new insight into the poorly understood mechanisms by which CDK-Mediator is targeted to genes and functions to control gene expression. We believe these important points are now more clearly framed in the revised manuscript.

6) The conclusion that FBXL19 does not appear to regulate CDK8-Mediator levels is not fully supported by the data. Figure 2—figure supplement 2H needs to be repeated so that CDK8 bands are not over-exposed as there does appear to be a change in amount/mobility when overexpressed -/+ MG132. Furthermore, there seems to be a noticeable increase of MED12 protein upon FBXL19 overexpression, which was even more obvious when cells were treated with MG132.

We have repeated these experiments based on the reviewer’s suggestions. We again failed to observe any significant difference in CDK8 protein levels following FBXL19 over-expression at lower exposure levels. We agree it appeared that MED12 levels may have increased slightly when FBXL19 was overexpressed in original Figure 2—figure supplement 1G. However, we believe this is an artefact resulting from inefficient transfer on the left had side of the gel. MED12 is a large protein (250KDa) and we previously have encountered difficulties efficiently transferring MED12 to membrane for Western blot analysis. We have now optimised our transfer of large proteins and fail to see any significant or reproducible alterations in MED12 levels (Figure 2—figure supplement 1H). MED12 is known to be regulated by the proteasome in ES cells (e.g. Buckley et al., 2012). In agreement with this, we observe an increase in MED12 levels after MG132 treatment in both FBXL19 overexpressing cells and in the control cells transfected with an empty vector. This effect is therefore not related to FBXL19. Finally, we have also included in the revised manuscript a Western blot analysis of MED13 (which is also part of the CDK module) and again we observe no change in protein levels following FBLX19 overexpression (Figure 2—figure supplement 1H). We conclude that FBXL19 does not regulate CDK module protein levels.

7) In Figure 4, CDK8 recruitment has only been tested for the FBXL19 ZF-CxxC deletion -is this recapitulated for depletion of the whole protein? Given that Figure 4B appears to show a decrease in the levels of the deletion protein this is an important point to verify.

We reasoned that conditionally removing the ZF-CxxC domain would be the most surgical way of disrupting binding of FBXL19 to CpG islands, without interfering other potential functions of FBXL19. We agree that deletion of the ZF-CxxC domain appears to cause some reduction in FBXL19 protein levels. We speculate that FBXL19 may be more rapidly degraded if not bound to chromatin (something we have observed for other ZF-CxxC domain containing proteins). Importantly, CDK8 proteins levels are unaffected in the ΔCxxC-FBXL19 cells (Figure 4—figure supplement 1F). We have now carried out a biochemical purification of ΔCxxC-FBXL19 and shown that it still interacts with CDK8-Mediator (Figure 2—figure supplement 1E). Given that ΔCxxC-FBXL19 does not bind to CpG islands (Figure 1) yet still interacts with CDK8, we conclude that reductions in CDK8 recruitment at FBXL19 targets genes is most likely explained by the loss of FBXL19 binding to CpG islands. Nevertheless, we now draw attention to the reduced protein levels of the ZF-CxxC deleted protein as follows (subsection “FBXL19 is required for appropriate CDK8 occupancy at a subset of CpG island-associated promoters”):

‘Following removal of the ZF-CxxC domain of FBXL19, we observed some reductions in FBXL19 protein levels (Figure 4B), but importantly CDK8 levels were unaffected (Figure 4—figure supplement 1F).’

8) The authors need to show that OHT treatment of the delCxxC ES line leads to reduced binding of FBXL19 in conditions used for CDK8 ChIPseq -several genes would be ok -these same genes should be used to validate CDK8 CHIPseq signal.

All commercially available antibodies for FBXL19 that we have tried fail to ChIP FBXL19 and our exhaustive attempts to generate our own FBXL19-specific antibody that works for ChIP have been unsuccessful (two separate epitopes and 4 rabbits). This has precluded us from carrying out the proposed experiment. However, we do observe a uniform loss of FBXL19 binding when the ZF-CxxC domain is removed in epitope-tagged transgene experiments (Figure 1) suggesting that FBXL19 binding to CpG islands relies on an intact ZF-CxxC domain.

9) Supplemental tables with differential gene expression results should be provided -or at least working gene lists used to reach the conclusions in the manuscript. Full GO enrichment results should be included as supplementary tables so that specific gene lists can be examined.

This information is now included as Supplementary file 1.

10) Various figure panels are missing statistical tests: Figure 4C needs statistical tests, e.g. KS/Wilcoxon -See Figure 4E; Figure 4G needs statistical tests; Figure 4H and 4I need boxplots and need statistical tests eg KS/Wilcoxon as for E/F;

We have now included the necessary statistical tests and plots as requested by the reviewer.

Figure 4H-J should include FBXL19-bound groups;

As suggested by the reviewer, we divided all CDK8 peaks into those which have FBXL19 (FBXL19+) and those that do not have FBXL19 (Fbxl19-). We then examined FBXL19, BioCAP and CDK8 enrichment in these groups (Figure 4—figure supplement 1I). As observed when we analysed all CDK8 peaks (Figure 4H-J), we found that CDK8 peaks that are bound by FBXL19 have increased levels of FBXL19, CDK8, and BioCAP signal when compared to sites not bound by FBXL19. Furthermore, sites that rely on FBXL19 for CDK8 binding have broader peaks of FBXL19 enrichment. This is consistent with the fact that these are large CpG islands with broad BioCAP signal (Figure 4F, 4H). The majority peaks that show reduction of CDK8 binding in FBXL19^ΔCXXC^ ES cells with FBXL19 peaks (519/783) while there is only a small overlap between FBXL19 binding and sites with increased CDK8 binding (90/379). We have now included this information in Figure 4—figure supplement 1I.

Figure 6B,E, F need statistical tests.

We have now included the necessary statistical tests.

11) It is very nice that the authors showed that genome-wide chromatin binding of FBXL19 was completely lost when the ZF-CxxC domain was deleted or mutated. To further confirm that it is indeed this ZF-CxxC domain mediating interaction with the non-methylated CpG islands, some controls are necessary. For example, in Figure 1—figure supplement 1D, the authors showed that almost all of the FBXL19 binding sites are overlapped with NMIs locations, with only 93 exceptions. Then are the signals at these 93 sites also changed in the FBXL19 mutant ChIPSeq data? When the cell DNA methylation levels are modulated, will the binding pattern of FBXL19 changed accordingly?

We have now compared the enrichment of WT FBXL19 and ΔCxxC FBXL19 at these 93 sites (Author response image 3), which corresponded to regions that were not associated with NMIs and observed a loss of binding of ΔCxxC FBXL19 (Author response image 3). Importantly, these same 93 sites also show some BioCAP signal, albeit at a level much lower than is observed at NMIs. This suggests that these sites are very weak CpG islands which were not captured in our NMI peak set due to their low Bio-CAP signal.

Furthermore, to illustrate the relationship between FBXL19 binding and CpG-rich non-methylated DNA, we have examined BioCAP and FBXL19 signal across all ATAC-seq peaks in the mouse ES cell genome (Author response image 3). ATAC-seq peaks represent a combination of regulatory elements that are CpGrich (CpG islands) or CpG-poor (usually distal regulatory elements). This provides the necessary contrast to illustrate the correlation of FBXL19 tracks with sites that contain CpG rich non-methylated DNA (BioCAP), consistent with a FBXL19 binding modality that is dependent on its ZF-CxxC domain.

We have not been able to examine FBXL19 binding in cells with altered DNA methylation levels as this would require recreation of a DNMT1 (or DNMT3A/B) null situation in our transgene containing cell line (we don’t use 5aza-C for this purpose as it causes extensive DNA damage due to covalent adducts which are formed between DNA and DNMT enzymes). However, when we have previously examined DNA methylation-deficient cells and the behaviour of the closely related KDM2A/B proteins, which have nearly identical CpG island binding patterns to FBXL19 (Figure 1D and 1E), we demonstrated that KDM2A/B occupancy is completely dependent on recognition of CpG-rich non-methylated DNA (Blackledge et al., 2010; Zhou et al., 2012).

**Author response image 3. respfig3:** (A) A Venn diagram showing the overlap between FBXL19 peaks and NMIs; (B) Metaplot analysis of enrichment of WT FS2-FBXL19, ΔCxxC FS2-FBXL19, and EV control at FBXL19-occupied NMIs (solid line) and nonNMI (dashed line) regions; (C) Heatmaps showing enrichment of BioCAP, WT FS2-FBXL19, ΔCxxC FS2FBXL19 and EV control at DNA accessible regions measured by ATAC-seq.

[Editors' note: the author responses to the re-review follow.]

Essential revisions:1) In AP-MS (Figure 2A), what is the background value of emPAI? Are there any other FBXL19-interacting proteins that show significantly high levels of emPAI in the list? How significant are the components of CDK-Mediator complex represented in all the hits?

To clarify these points, we have now included a table in the supplementary information (Figure 2—source data 1) detailing the AP-MS results. This includes the biological replicate FS2-FBXL19 purifications and matched empty vector (EV) controls. We limited analysis of our AP-MS data to proteins that were uniquely identified in the FBXL19 purifications (i.e. not the EV control). We have included a column highlighting the proteins that were identified in this screen and their associated emPAI scores in Supplementary file 1. Importantly, Mediator components feature heavily in this table, which is what prompted us to examine CDK-Mediator as a possible interaction of functional relevance. We have yet to validate whether other proteins in this table are bona fide interactors and, therefore, their relationship with FBXL19 and its function is not known.

2) In Figure 3, the authors demonstrated that FBXL19 was able to recruit CDK-Mediator complex, such as CDK8 and MED12, to artificially designed chromatin regions. What about CXXC-domain or F-box deletion mutants in this assay?

In these experiments we purposely used a version of FBXL19 in which the CxxC domain has a single substitution mutation (K49A) that prevents CpG island binding. We designed the experiment in this way to prevent the TetR-fusion protein from binding to CpG islands throughout the genome. This allowed us to focus our analysis on the artificial nucleation site. Although we have not tested a version of FBXL19 with its CxxC domain deleted, we believe that it would function in a very similar way to the K49A mutant as both proteins are deficient in CpG island binding and we have shown that ΔCxxC-FBXL19 retains its interaction with CDK-Mediator (Figure 2—figure supplement 1E).

In all of our artificial tethering experiments we generate clonal stable cell lines expressing the TetR-fusion proteins to ensure uniform expression within lines and comparable expression across test lines. Despite exhaustive efforts, we have been unable to generate stable cell lines expressing FBXL19 lacking the F-box. We are unsure why this is the case but speculate that FBXL19 without the F-box may be unstable or lead to a dominant negative influence that affects cell growth (negative selection). Therefore, we have not been able to examine how the F-box deletion behaves in our tethering experiments. Given that ΔF-box-FBXL19 does not interact with CDK-Mediator in transient overexpression IP experiments (Figure 2—figure supplement 1E), we assume that CDK8 and MED12 would not be recruited to chromatin by TetR-ΔF-box-FBXL19. However, we have been unable to directly test this possibility.

3) In Figure 4B, it is not very clear which band(s) are exactly indicated as full-length FBXL19 and which one(s) are CXXC deletion? Why are there double bands? Did the author use T7 antibody to detect the endogenously introduced protein?

We generated the T7 knock-in FBXL19 ES cell line (Figure 2—figure supplement 1B, C) in order to detect FBXL19 by Western blot. We observe ‘double bands’ for FBXL19 and speculate that this may be due to post-translational modification. The closely related KDM2A and KDM2B proteins are also extensively post-translationally modified leading to doublets when examined by Western blot. Following tamoxifen treatment of *Fbxl19^CxxCfl/fl^* cells to remove the CxxC domain, we observe a reduction in mobility for these double bands (with the top band shifting to the previous position of the bottom band). We have now clearly indicated which bands represent WT and ΔCxxC FBXL19 in Figure 4B and have described this in the figure legend.

4) In Figure 4C, what proportions of each group of CDK8 binding sites overlap with FBXL19? In theory, the author expects to see that CDK8 sites showing binding intensity changes upon tamoxifen treatment would tend to overlap more with FBXL19.

The overlap of each group of CDK8 peaks with FBXL19 peaks is shown in Figure 4—figure supplement 1H. While roughly 41% of all CDK8 peaks overlap with FBXL19 peaks, the overlap of down-regulated CDK8 peaks is about 66%. This is highlighted in the text in subsection “FBXL19 is required for appropriate CDK8 occupancy at a subset of CpG island-associated promoters”. On the other hand, we observed that only 23% of up-regulated CDK8 peaks overlap with FBXL19 peaks.

5) In Figure 4—figure supplement 1C, please use a Venn Diagram to show directly the number of peaks of CDK8, FBXI19 and NMI overlapping with each other. This would be more straightforward.

We chose to represent CDK8 enrichment at FBXL19 peaks as a heatmap in order to circumvent the limitation of peak-calling. However, as suggested by the reviewer, we have now also included a Venn diagram that shows that 89.4% of FBXL19 peaks overlap with CDK8 peaks and NMIs (Figure 4—figure supplement 1D) and mentioned this in the text (subsection “FBXL19 is required for appropriate CDK8 occupancy at a subset of CpG island-associated promoters”).

6 In Figure 5, please perform the same GSEA or functional annotation analysis on genes that are associated with unchanged CDK8 binding sites upon FBXL19 CXXC-domain deletion.

We have now included the GO analysis for genes associated with unchanged CDK8 binding in Figure 5—figure supplement 1. This revealed a broad range of terms associated with basic molecular processes in line with Mediator being implicated as a key regulator of transcription. We now highlight this observation in the text in subsection “FBXL19 targets CDK8 to promoters of silent developmental genes in ES cells”.

7) In Figure 6B, what is the result for genes that are associated with increased CDK8 binding signals in FBXL19 CXXC-domain deletion mutant upon RA-induced differentiation?

We have compared the expression of genes associated with an increase in CDK8 binding in Figure 6—figure supplement 1E. While we observe no overall difference in the expression of these genes in ES cells, there is a slight increase upon RA-induced differentiation (p-value 0.04). This is consistent with Mediator playing a role in gene transcription.

8) In Figure 6C, even though the authors claimed that those upregulated genes (889) in FBXL19 CXXC-domain deletion mutant upon RA treatment were not involved in developmental processes, it is still interesting to find that significantly more genes were activated upon CXXC-domain deletion. What explanation can the authors provide for this finding?

We agree with the reviewer that a significant number of genes were more activated in the *Fbxl19^ΔCxxC^* cells compared to wild type cells upon RA-induced differentiation and that this is potentially interesting. We have yet to find a compelling molecular explanation to link FBXL19 directly to these effects on gene expression. It remains possible that these gene expression changes are due to secondary transcriptional defects that manifest when cells attempt to execute a new expression programme in the absence of normal activation of silent, lineage commitment-associated genes when FBXL19 is not functional, or that FBXL19 plays other, as yet unidentified, roles in inhibiting gene expression. We have now added a sentence drawing attention to these points in the revised manuscript in subsection “Removing the CpG island-binding domain of FBXL19 results in a failure to induce developmental genes during ES cell differentiation”.

9) In Figure 7, the authors claim that FBXL19 relies on CDK-mediator complex to activate the specific set of developmental genes. Is it possible to test this hypothesis in the FBXL19 CXXC-deletion system they established in Figure 6? In wild-type ESC, depletion of MED13 will have similar effects as FBXL19 CXXC deletion, whereas in FBXL19 mutant, the effects would be much alleviated?

One could test whether inducible MED13/13L deletion in the *Fbxl19^CxxCfl/fl^* background would not lead to any further effect on gene activation. However, this would require a significant amount of time to generate and characterise a new *Med13^fl/fl^, Med13l^fl/fl^*, and *Fbxl19^CxxCfl/fl^*cell line. While this is an interesting experiment, we believe that the robust and extensive series of observations in Figure 6 and Figure 7 more than adequately support our conclusion that FBXL19 target genes rely on CDK-Mediator for activation during differentiation.

10) In Figure 8, why did heterozygous deletion of FBXL19 CXXC-domain show even better developmental status than the wild type?

We have not observed any significant difference between WT and heterozygous mutant embryos. It is common to have some stage variations until around 12 dpc, even within embryos of the same litter. Therefore, we often find small differences in size and developmental status among embryos regardless of genotypes. Both the WT and heterozygous mutant embryos presented in Figure 8 appear healthy with normal development for this stage. On the other hand, the defects observed in the homozygous *Fbxl19^ΔCxxC^* embryos cannot be explained by such stage variations.

11) The authors demonstrated that the CXXC-domain mediates the binding of FBXL19 at NMIs, where F-Box mediates its interaction with the CDK-Mediator complex. What about CDK8 chromatin binding or gene expression patterns upon F-Box deletion of FBXL19? Will the authors see the same results as what they found in CXXC-domain deletion mutant?

This is an interesting experiment and we hypothesize that disrupting the interaction between CDK-Mediator and FBXL19 via deleting the F-box domain would yield similar results to removing the CxxC domain. However, as described above in response to reviewer point 2, despite exhaustive efforts we have been unable to generate cell lines stably expressing FBXL19 with a deletion of the F-box, precluding us from addressing this point. It will be interesting in future work to see if we can engineer a cell line where we are able to remove the F-box in an inducible manner and examine the effects on CDK8 binding and induction of gene expression.